A little frog leaps a long way: compounded colonizations of the Indian Subcontinent discovered in the tiny Oriental frog genus Microhyla (Amphibia: Microhylidae)

Gorin Vladislav A. 1
Solovyeva Evgeniya N. 2
Hasan Mahmudul 3
Okamiya Hisanori 4
http://orcid.org/0000-0003-0965-7781 Karunarathna D.M.S. Suranjan 5
Pawangkhanant Parinya 6
de Silva Anslem 7
Juthong Watinee 8
Milto Konstantin D. 9
Nguyen Luan Thanh 10
Suwannapoom Chatmongkon 6
http://orcid.org/0000-0002-3961-518X Haas Alexander 11
http://orcid.org/0000-0003-1478-3387 Bickford David P. 12
Das Indraneil 13
http://orcid.org/0000-0002-7576-2283 Poyarkov Nikolay A. 1 14 n.poyarkov@gmail.com
1 Faculty of Biology, Department of Vertebrate Zoology, Lomonosov Moscow State University , Moscow , Russia
2 Zoological Museum, Lomonosov Moscow State University , Moscow , Russia
3 Department of Fisheries, Bangamata Sheikh Fazilatunnesa Mujib Science & Technology University , Jamalpur , Bangladesh
4 Department of Biological Science, Faculty of Science, Tokyo Metropolitan University , Tokyo , Japan
5 Nature Explorations and Education Team , Moratuwa , Sri Lanka
6 School of Agriculture and Natural Resources, University of Phayao , Phayao , Thailand
7 Amphibia and Reptile Research Organization of Sri Lanka , Gampola , Sri Lanka
8 Prince of Songkla University , Songkhla , Thailand
9 Zoological Institute, Russian Academy of Sciences , St. Petersburg , Russia
10 Asian Turtle Program—Indo-Myanmar Conservation , Hanoi , Vietnam
11 Center for Natural History, Universität Hamburg , Hamburg , Germany
12 Biology Department, University of La Verne , La Verne, CA , USA
13 Institute of Biodiversity and Environmental Conservation, Universiti Malaysia Sarawak , Kota Samarahan , Malaysia
14 Joint Russian-Vietnamese Tropical Research and Technological Center , Hanoi , Vietnam
Measey John
Electronic publication date: 2020 Jul 3
Publication date: 2020
Volume: 8
Electronic Location ID: e9411
Received 2020 Feb 27; Accepted 2020 Jun 3
Copyright: © 2020 Gorin et al.
Copyright year: 2020
Copyright holder: Gorin et al.
License: This is an open access article distributed under the terms of the Creative Commons Attribution License, which permits unrestricted use, distribution, reproduction and adaptation in any medium and for any purpose provided that it is properly attributed. For attribution, the original author(s), title, publication source (PeerJ) and either DOI or URL of the article must be cited.
License URL: https://creativecommons.org/licenses/by/4.0/

Keywords: Molecular phylogeny, Biogeography, Miniaturization, Narrow-mouthed frogs, Southeast Asia, Microhylinae, Species delimitation, Indian collision, Cryptic species, Glyphoglossus

Funding: Russian Science Foundation 19-14-00050 Thailand Research Fund DBG6180001 Biodiversity and Natural Resources Management, University of Phayao UoE63005 Rufford Foundation 23951-1 Niche Research Grant Scheme, Ministry of Higher Education, Government of Malaysia NRGS/1087/2013(01) Russian-Vietnamese Tropical and Technological Center (JRVTTC) This work was supported by the Russian Science Foundation to Nikolay A. Poyarkov (RSF grant No. 19-14-00050; specimen collection, molecular, phylogenetic and morphological analyses, data analysis); by the Thailand Research Fund (DBG6180001) and the Unit of Excellence 2020 on Biodiversity and Natural Resources Management, University of Phayao (UoE63005) to Chatmongkon Suwannapoom (specimen collection); by the Rufford Foundation (23951-1; sampling) to Suranjan Karunarathna; and by the Niche Research Grant Scheme, Ministry of Higher Education, Government of Malaysia (NRGS/1087/2013(01); partial sampling, molecular analysis) to Indraneil Das. Fieldwork in Vietnam was funded by the Joint Russian-Vietnamese Tropical and Technological Center (JRVTTC). The funders had no role in study design, data collection and analysis, decision to publish, or preparation of the manuscript.

==============================
Frogs of the genus Microhyla include some of the world’s smallest amphibians and represent the largest radiation of Asian microhylids, currently encompassing 50 species, distributed across the Oriental biogeographic region. The genus Microhyla remains one of the taxonomically most challenging groups of Asian frogs and was found to be paraphyletic with respect to large-sized fossorial Glyphoglossus. In this study we present a time-calibrated phylogeny for frogs in the genus Microhyla, and discuss taxonomy, historical biogeography, and morphological evolution of these frogs. Our updated phylogeny of the genus with nearly complete taxon sampling includes 48 nominal Microhyla species and several undescribed candidate species. Phylogenetic analyses of 3,207 bp of combined mtDNA and nuDNA data recovered three well-supported groups: the Glyphoglossus clade, Southeast Asian Microhyla II clade (includes M. annectens species group), and a diverse Microhyla I clade including all other species. Within the largest major clade of Microhyla are seven well-supported subclades that we identify as the M. achatina, M. fissipes, M. berdmorei, M. superciliaris, M. ornata, M. butleri, and M. palmipes species groups. The phylogenetic position of 12 poorly known Microhyla species is clarified for the first time. These phylogenetic results, along with molecular clock and ancestral area analyses, show the Microhyla—Glyphoglossus assemblage to have originated in Southeast Asia in the middle Eocene just after the first hypothesized land connections between the Indian Plate and the Asian mainland. While Glyphoglossus and Microhyla II remained within their ancestral ranges, Microhyla I expanded its distribution generally east to west, colonizing and diversifying through the Cenozoic. The Indian Subcontinent was colonized by members of five Microhyla species groups independently, starting with the end Oligocene—early Miocene that coincides with an onset of seasonally dry climates in South Asia. Body size evolution modeling suggests that four groups of Microhyla have independently achieved extreme miniaturization with adult body size below 15 mm. Three of the five smallest Microhyla species are obligate phytotelm-breeders and we argue that their peculiar reproductive biology may be a factor involved in miniaturization. Body size increases in Microhyla—Glyphoglossus seem to be associated with a burrowing adaptation to seasonally dry habitats. Species delimitation analyses suggest a vast underestimation of species richness and diversity in Microhyla and reveal 15–33 undescribed species. We revalidate M. nepenthicola, synonymize M. pulverata with M. marmorata, and provide insights on taxonomic statuses of a number of poorly known species. Further integrative studies, combining evidence from phylogeny, morphology, advertisement calls, and behavior will result in a better systematic understanding of this morphologically cryptic radiation of Asian frogs.

Introduction

The tropical areas of South and Southeast Asia include biogeographic regions recognized as global centers of biodiversity (Myers et al., 2000; Bain et al., 2008; Stuart, 2008; De Bruyn et al., 2014). Understanding processes that sculpted this diversity is hampered by a highly complex geological and climatic history of this region. Combining data on tectonics, paleoclimate, and phylogenetics has proved to be a powerful instrument for examining patterns of diversification within clades and understanding processes involved in the assembly of high biodiversity in regions like South and Southeast Asia (De Bruyn et al., 2014).

The tectonic collision between the Indian subcontinent (ISC) and the Eurasian landmass during the Early Cenozoic is widely recognized as a key event that caused significant geologic and climatic changes, such as the rise of the Himalaya, uplift of the Tibetan plateau, and a general drying of Central Asia (Harrison et al., 1992; An et al., 2001; Guo et al., 2002; Molnar, 2005; Solovyeva et al., 2018). This tectonic event also induced a major biotic interchange between the ISC and Eurasia and is widely regarded as a major driver of biotic diversification (Wilkinson et al., 2002; Roelants, Jiang & Bossuyt, 2004; Li et al., 2013; Garg & Biju, 2019). Numerous studies have demonstrated that floral and faunal elements reached and colonized tropical Asia from Gondwanaland via the northward drifting ISC (Dayanandan et al., 1999; Klaus et al., 2010, 2016; Kamei et al., 2012; Morley, 2018), the so called “out-of-India” hypothesis (Bossuyt & Milinkovitch, 2001; Conti et al., 2002; Gower et al., 2002; Biju & Bossuyt, 2003; Sparks, 2003; Dutta et al., 2004; Karanth, 2006; Datta-Roy & Karanth, 2009). At the same time, a set of phylogenetic studies of different animal groups proposed an alternative “out-of-Eurasia” biogeographic hypothesis, suggesting a Southeast Asian origin of these lineages with further dispersal and colonization of the ISC during its collision with the Eurasian landmass (Raxworthy, Forstner & Nussbaum, 2002; Renner, 2004; Köhler & Glaubrecht, 2007; Van der Meijden et al., 2007; Macey et al., 2008; Grismer et al., 2016; Garg & Biju, 2019).

There is an ongoing debate on timing and topography of the ISC–Eurasian collision (Acton, 1999; Van Hinsbergen et al., 2011, 2012; Ali & Aitchison, 2008, 2012). Some recent geologic models suggest land bridges connected ISC and modern Southeast Asia since the early Eocene (ca. 55–35 MYA), well before collision of the Indian plate with Eurasia (30–25 MYA) (Aitchison, Ali & Davis, 2007; Aitchison & Ali, 2012; Hall, 2012; Ding et al., 2017). Several phylogenetic studies corroborate the existence of pre-collision faunal exchanges between the ISC and Southeast Asia, demonstrating that they likely went in both directions: “out-of-India” and “out-of-Eurasia” (Klaus et al., 2010; Li et al., 2013; Grismer et al., 2016; Garg & Biju, 2019). Overall, the impact of the “out-of-India” or “out-of-Eurasia” biogeographic scenarios in pre-collision or post-collision biotic exchanges between the ISC and the Asian mainland remains insufficiently studied and unresolved.

Frogs of the family Microhylidae are some of the most species rich groups of Anura, comprising 690 species in 12 subfamilies (Frost, 2020; Streicher et al., 2020). Because of their transcontinental circumtropical distribution, microhylids are a promising test case for biogeography studies (Savage, 1973; Van Bocxlaer et al., 2006; Van der Meijden et al., 2007; Kurabayashi et al., 2011). Among the 12 currently recognized subfamilies of microhylids, the subfamily Microhylinae is widely distributed in South, Southeast, and East Asia currently including eight genera with nearly one hundred species (Garg & Biju, 2019; Frost, 2020). Their phylogenetic relationships and historical biogeography have been discussed in several studies (Van Bocxlaer et al., 2006; Frost et al., 2006; Matsui et al., 2011; De Sá et al., 2012; Peloso et al., 2016; Feng et al., 2017). The most recent analysis of genus-level phylogeny within the Microhylinae by Garg & Biju (2019) suggested their origin on the ISC during early Paleocene with dispersal to the Asian mainland via several Eocene land bridges connecting the ISC with Southeast Asia. Following accretion of India and Eurasia in the Oligocene/Miocene, some Microhylinae lineages that diversified in Southeast Asia could have recolonized the ISC (Garg & Biju, 2019). However, phylogenetic relationships and historical biogeography within Microhylinae genera remain poorly resolved.

The genus Microhyla is the most species rich genus in the Microhylinae, currently comprising 50 recognized species (Poyarkov et al., 2014, 2019; Biju et al., 2019). Over half of this diversity has been described within the last 15 years (27 species, see Frost, 2020), yet Microhyla remains one of the most taxonomically challenging groups of Asian frogs. Most species of Microhyla are small-sized terrestrial frogs, while several diminutive species approach the lower body-size limit for vertebrates and represent some of the world’s tiniest amphibians (Das & Haas, 2010; Poyarkov et al., 2014). In miniaturized groups of amphibians, a high proportion of cryptic diversity and rampant homoplasies are often recorded, obscuring our estimates of diversity and evolutionary relationships (Hanken & Wake, 1993; Rovito et al., 2013; Parra-Olea et al., 2016; Rakotoarison et al., 2017). Molecular phylogenetic analyses, optimally combined with behavioral and acoustic data, offer the best hope for clarifying diversity, species boundaries, and evolutionary relationships in many groups of Microhylidae, including the genus Microhyla (Hasan et al., 2014a; Garg et al., 2019; Poyarkov et al., 2018a, 2019).

Despite significant progress in our understanding of Microhyla diversity in recent years, hypothesizing evolutionary origins of the genus remains a challenging task. Microhyla is the only Asian microhylid genus with a wide distribution over South, Southeast, and East Asia (see Fig. 1), making it an ideal model for studies on Asian biogeography. Phylogenetic relationships among members of the genus Microhyla have been discussed in several recent studies (Matsui et al., 2011; Peloso et al., 2016; Tu et al., 2018; Nguyen et al., 2019; Biju et al., 2019); however, they were generally based on limited sampling (<60% of recognized diversity). Monophyly of Microhyla was questioned by Matsui et al. (2011), based on analysis of mitochondrial DNA (mtDNA) markers, but later corroborated by multi-locus phylogenetic analyses (Peloso et al., 2016; Tu et al., 2018). Only a few works have provided insights on biogeographic origin, patterns of distribution, and possible routes of colonization for the genus (Vineeth et al., 2018; Garg et al., 2019; Poyarkov et al., 2019). Though the greatest species diversity of Microhyla is observed in Southeast Asia (up to nine sympatric species in Indochina, see Fig. 1), some studies suggested the possibility of an Indian origin for the genus and several species groups (Garg et al., 2019; Garg & Biju, 2019). That may be explained by biased taxonomic and geographic representation of Microhyla species in these works, primarily focused on South Asian taxa. Extensive taxon sampling of all known members of the genus Microhyla, including molecularly previously unstudied taxa from Southeast Asia, a more robustly resolved phylogeny, and sound divergence age estimates are important to understand historical distribution and diversification in this radiation of Asian frogs (Garg et al., 2019). Thus far, however, a comprehensive phylogenetic investigation with dating estimates within the genus Microhyla is lacking.

Figure 1 Distribution and species richness of Oriental tiny frogs of the genus Microhyla.

Heatmap indicates approximate number of sympatrically co-occurring species (from 1 to 9); the highest species richness is observed in southern Vietnam and Malayan Peninsula. The individual species geographic range maps were adopted from the AmphibiaWeb (2020) database, and modified based on expert estimations in CorelDraw Graphics Suite X8. Grey/white dashed lines mark international borders on land/water, respectively. Base Map created using simplemappr.net. Photo shows Microhyla heymonsi—a widespread species occurring in Southeast and East Asia (by Nikolay A. Poyarkov).

Herein, we identify unrecognized diversity and examine phylogenetic relationships among almost all described species of Microhyla based on extensive geographic and taxonomic sampling, including 48 of 50 nominal species (96% of recognized taxa); phylogenetic information for 12 species and a number of candidate new species is reported for the first time. We use the resulting phylogeny (based on both mtDNA and nuDNA markers) to test biogeographic hypotheses in space and time and provide a scenario for Microhyla diversification. Our study provides the first nearly complete phylogeny for the genus Microhyla and links the Indian Subcontinent with Sundaland, Indo-Burma, and East Asia, thereby allowing a better understanding of biogeographic history and diversification of the group. We also evaluate miniaturization and simulate body size evolution across different lineages of the genus, providing novel insights into morphological evolution in Microhyla.

Materials and Methods

Taxon sampling

We used tissues from the herpetological collections of Zoological Museum of Moscow University (ZMMU; Moscow, Russia); Zoological Institute, Russian Academy of Sciences (ZISP; St. Petersburg, Russia); Vertebrate Zoology Department, Biological Faculty, Moscow State University (ZPMSU; Moscow, Russia); Amphibian Research Center, Hiroshima University (IABHU; Higashihiroshima, Japan); Danum Valley Conservation Area, Specimen collection (RMBR; Sabah, Malaysia); Department of Fisheries, Bangamata Sheikh Fazilatunnesa Mujib Science & Technology University (DFBSFMSTU; Jamalpur, Bangladesh); School of Agriculture and Natural Resources, University of Phayao (AUP; Phayao, Thailand); and the Institute of Biodiversity and Environmental Conservation, Universiti Malaysia Sarawak (UNIMAS; Sarawak, Malaysia) (information summarized in Table S1). Permissions to conduct fieldwork and collect specimens were granted by the Institutional Ethical Committee of Animal Experimentation of University of Phayao (permit number 610104022), the Institute of Animals for Scientific Purpose Development (IAD), Bangkok, Thailand (permit number U1-01205-2558), the Sarawak Forest Department and the Sarawak Forestry Corporation (permit number JHS/NCCD/600-7/2/107(Jld2)), the Department of Wildlife Conservation of Sri Lanka (permit number WL/3/2/1/14/12), the Forest Protection Departments of the Peoples’ Committee of Gia Lai Province (permit number #530/UBND-NC), the Department of Forestry, Ministry of Agriculture and Rural Development of Vietnam (permit numbers #142/SNgV-VP, #1539/TCLN-DDPH, #1700/UBND.VX and #308/SNgV-LS).

We analyzed 122 tissue samples representing 40 nominal species of Microhyla, 14 species have not been included in previous phylogenetic analyses. In our analysis, we also included GenBank sequences from 78 specimens of approximately 37 nominal Microhyla species, 29 other Microhylidae representatives, and five non-microhylid outgroups used for rooting the phylogenetic tree and divergence times estimation (Table S1). In total, we obtained molecular genetic data for 199 samples representing 48 nominal Microhyla species. Geographic location of sampled populations is presented in Fig. S1. For alcohol-preserved voucher specimens stored in museum collections, we removed a small sub-sample of muscle, preserved it in 96% ethanol, and stored samples at −70 °C.

DNA extraction, amplification and sequencing

For molecular phylogenetic analyses, total genomic DNA was extracted from ethanol-preserved femoral muscle tissue using standard phenol-chloroform-proteinase K (final concentration 1 mg/ml) extraction procedures with consequent isopropanol precipitation (protocols followed Russell & Sambrook, 2001).

For mtDNA, we amplified sequences covering fragments of 12S rRNA, tRNAVal, and 16S rRNA mtDNA genes to obtain an up to 2478 bp-length continuous fragment of mtDNA. The 16S rRNA gene has been widely applied in biodiversity surveys in amphibians (Vences et al., 2005; Vieites et al., 2009) and 12S rRNA + 16S rRNA data have been used in several important studies on Microhylinae phylogeny (Matsui et al., 2011; Peloso et al., 2016). These fragments have also proven to be particularly useful in taxonomic studies of the genus Microhyla and closely-related taxa (Hasan et al., 2012, 2014a, 2014b; Howlader et al., 2015; Matsui, 2011; Matsui et al., 2011; Matsui, Hamidy & Eto, 2013; Wijayathilaka et al., 2016).

For nuDNA, we amplified a 729 bp-long fragment of brain-derived neurotrophic factor gene (BDNF). This marker was recently successfully applied in biodiversity and phylogenetic studies of Indian Microhyla species (see Garg et al., 2019; Garg & Biju, 2019; Biju et al., 2019). Primers used in PCR and sequencing were taken from the literature or designed by us and summarized in Table S2.

The PCR conditions for amplifying mtDNA fragments included an initial denaturation step of 5 min at 94 °C, and 40 cycles of denaturation for 1 min at 94 °C, primer annealing step for 1 min with TouchDown program from 65 °C reducing 1 °C every cycle to 55 °C, and extension step for 1 min at 72 °C, and the final extension step for 5 min at 72 °C. The PCR conditions for amplifying BDNF gene followed Van der Meijden et al. (2007) and included an initial denaturation step of 5 min at 94 °C followed with 32 cycles of denaturation for 1 min at 94 °C, primer annealing step for 1 min at 50 °C, and extension for 1 min at 72 °C, with final extension step for 5 min at 72 °C. PCRs were run on an Bio-Rad T100™ Thermal Cycler; sequence data collection and visualization were performed on an ABI 3730xl automated sequencer (Applied Biosystems, Foster City, CA, USA). PCR purification and cycle sequencing were done commercially through Evrogen® (Moscow, Russia). All unique sequences were deposited in GenBank (see Table S1).

Phylogenetic analyses

In addition to newly obtained sequences, we also used 107 DNA sequences of Microhylidae from GenBank in our final alignments; sequences of Rhacophorus schlegelii, Alytes dickhilleni, A. muletensis, Blommersia transmarina and B. wittei were selected as outgroup taxa to help root our tree and were also used for time-tree calibration. Details on taxonomy, localities, GenBank accession numbers, and associated references for all examined specimens are summarized in Table S1.

Nucleotide sequences were initially aligned in MAFFT v6 (Katoh et al., 2002) with default parameters, subsequently checked by eye in BioEdit v7.0.5.2 (Hall, 1999), and adjusted as needed.

We reconstructed phylogenetic trees with two datasets:

A mtDNA dataset joining 12S rRNA and 16S rRNA for all examined samples, used for assessment of species groups and estimation of cryptic diversity within Microhyla (230 sequences, including 199 sequences of Microhyla);

A concatenated mtDNA + nuDNA dataset, joining long 12S rRNA—16S rRNA mtDNA fragment and BDNF gene sequences for 118 selected samples representing all major lineages within Microhyla (as revealed by initial analysis of mtDNA), used for obtaining a more robust phylogenetic hypothesis, time-tree estimation, and ancestral range reconstruction for Microhyla.

The optimum partitioning schemes for alignments were identified with PartitionFinder 2.1.1 (Lanfear et al., 2012) using the greedy search algorithm under an AIC criterion, and are presented in Table S3. Phylogenies were hypothesized via maximum likelihood (ML) and Bayesian Inference (BI). We used IQ-TREE (Nguyen et al., 2015) to reconstruct ML phylogenies. Confidence in tree topology for ML analysis was assessed by 1,000 bootstrap replications for ML analysis (ML BS). Bayesian inference (BI) was performed in MrBayes v3.1.2 (Ronquist & Huelsenbeck, 2003) with two simultaneous runs, each with four chains for 200 million generations. We checked that the effective sample sizes (ESS) were all above 200 by exploring likelihood plots using TRACER v1.6 (Rambaut & Drummond, 2007). The initial 10% of trees were discarded as burn-in. Confidence in tree topology was assessed by posterior probability for Bayesian analysis (BI PP) (Huelsenbeck & Ronquist, 2001). We a priori regarded tree nodes with ML BS values 75% or greater and BI PP values over 0.95 as sufficiently resolved (Huelsenbeck & Hillis, 1993; Felsenstein, 2004). For clarity, ML BS values between 75% and 50% (BI PP between 0.95 and 0.90) were regarded as tendencies and below 50% (BI PP below 0.90) were considered unresolved. The allele network for the BDNF gene was constructed using median-joining method in the PopArt v1.5 (Leigh & Bryant, 2015) with 95% connection limit.

Species delimitation

We examined putative species boundaries beyond those currently recognized by taxonomists based on two different species delimitation methods: The Automatic Barcode Gap Discovery (ABGD; Puillandre et al., 2012) and the Generalized Mixed Yule-Coalescent model (GMYC; Pons et al., 2006). These methods enable the delimitation of independently-evolving species based on genetic data (Fujita et al., 2012; Dellicour & Flot, 2015; Eyer et al., 2017) and do not require a priori hypotheses of putative species groupings, thereby limiting potential bias in species delimitation.

The ABGD method is a single-gene approach to statistical detection of barcode gaps in a pairwise genetic distance distribution (Puillandre et al., 2012). Barcode gaps, presumably occurring between intra- and interspecific distances, were used to partition the 16S rRNA dataset into species hypotheses (initial partition). Resulting inferences were then recursively applied to yield finer partitions (recursive partitions) until no further partitioning was possible. ABGD analysis was run on the 16S rRNA dataset through a web-based interface (https://bioinfo.mnhn.fr/abi/public/abgd/) using default parameters (10 steps of intraspecific divergence prior from Pmin = 0.001 to Pmax = 0.10, X = 2).

The GMYC method is also a single-gene approach to identifying species “boundaries” associated with shifts in branching rates between intra- and interspecies branching events on a time-calibrated ultrametric tree (Pons et al., 2006; Fujisawa & Barraclough, 2013). We used a Bayesian implementation of this method (bGMYC; Reid & Carstens, 2012), which was applied to the 16S rRNA data. We obtained the distribution of ultrametric phylogenetic trees of 16S rRNA haplotypes with BEAST v1.8.4 (Drummond et al., 2012), then used 100 random phylogenetic trees as an input for subsequent bGMYC analysis. We ran bGMYC for 50,000 generations with burn-in 40,000 and a thinning parameter of 100. We summarized results of bGMYC analyses in a matrix of pairwise co-assignment probabilities for each haplotype, shown as a heatmap (not presented).

In addition, both inter- and intraspecific uncorrected genetic p-distances were calculated using MEGA 6.1 (Tamura et al., 2013).

Divergence times estimation

Molecular divergence dating was performed in BEAST v1.8.4, including the concatenated mtDNA + nuDNA dataset. We used hierarchical likelihood ratio tests in PAML v4.7 (Yang, 2007) to test molecular clock assumptions separately for mtDNA and nuDNA markers. Based on PAML results, we then decided to use a strict clock for the nuDNA (BDNF) and an uncorrelated lognormal relaxed clock for mtDNA. We also used these models and partitioning schemes from the ML analysis. The Yule model was set as the tree prior and we assumed a constant population size and default priors for all other parameters. We conducted two runs of 100 million generations each in BEAST v1.8.4. We also assumed parameter convergence in Tracer and discarded the first 10% of generations as burn-in. We used TreeAnnotator v1.8.0 (in BEAST) to create our maximum credibility clades.

Since we could find no paleontological data for the Microhylinae, we relied on three recently estimated calibration priors for this subfamily obtained from recent large-scale phylogenies of microhylids (Kurabayashi et al., 2011), and a fossil record of Gastrophryninae (Sanchiz, 1998; Holman, 2003; De Sá et al., 2012) as primary calibration points. We also applied two additional calibration points widely used in divergence time estimates of Anura: maximum age of the split between Blommersia wittei and B. transmarina from the Comoro islands at 15 MYA (Vences et al., 2003), and the minimum age of Alytes muletensis—A. dickhilleni split at 5 MYA (Fromhage, Vences & Veith, 2004). Calibration points and priors are summarized in Table S4.

Ancestral area reconstruction

To infer a biogeographic history of Microhyla, a model-testing approach was applied using the ML tree with Lagrange (Ree et al., 2005; Ree & Smith, 2008) in RASP v3.2 (Yu et al., 2015). Species occurrences were categorized according to nine biogeographic areas, modified from Turner, Hovenkamp & Van Welzen (2001), Wood et al. (2012) and Chen et al. (2018), reflecting patterns of endemism in Microhyla (see Fig. 2A): (A) Mainland East Asia; (B) Eastern Indochina; (C) Western Indochina; (D) Indian Subcontinent; (E) Malayan Peninsula; (F) Sumatra + Java + Bali; (G) Borneo and adjacent Philippine islands; (H) Sri Lanka; and (I) East Asian islands (Taiwan + the Ryukyus) (see Supplemental Information 1 for biogeographic area definitions and references). Maximum range-size was set to three areas, as no extant species occurs in more than three biogeographical regions. Matrices of modern distributions of taxa/ area are given in Table S5. We modeled discrete state transitions (for ranges) on branches as functions of time, enabling ML estimation of where ancestral linneages’ geographic areas were at the time of cladogensis. A Lagrange analysis found ancestral area(s) at a node, split areas into two distinct lineages, and calculated probabilities of most likely geographic areas for the nodes (Ree & Smith, 2008). Analyses used two models (Matzke, 2013): Langrange Dispersal-Extinction-Cladogenesis (DEC; Ree & Smith, 2008), and the ML version of Statistical Dispersal-Vicariance Analysis (S-DIVA; Ronquist, 1997). We assessed model fit using the Akaike Information Criterion (AIC) and Akaike weights.

Figure 2 Biogeographic history of Microhyla.

(A) Biogeographic regions used in the present study; (B) BEAST chronogram on the base of 3207 bp-long mtDNA + nuDNA dataset with the results of ancestral area reconstruction in RASP. For biogeographic areas definitions, species occurrence data and transition matrices see Supplemental Information 1, Tables S5 and S6. Information at tree tips corresponds to biogeographic area code (see Fig. 2A), sample number (summarized in Table S1), and species name, respectively. Node colors correspond to the respective biogeographic areas; values inside node icons correspond to node numbers (see Fig. S3 and Table S11 for divergence time estimates); values near nodes indicate marginal probabilities for ancestral ranges (S-DIVA analysis); icons illustrate vicariant and dispersal events (see Legend). Red arrows from 1 to 5 correspond to the dispersals to the Indian Subcontinent by Microhyla II lineages. Base Map created using simplemappr.net.

Given the tremendous geological complexity of the region through time, we applied the following time and dispersal constraints to the analyses: we set four periods, corresponding to the main stages of gradual ISC movements northwards, formation of the first land bridges between the ISC and Southeast Asia, and final accretion between the ISC and the Asian mainland based on data from recent geologic models (based on Hall, 2012; Ding et al., 2017; Morley, 2018; see Fig. 3 for schematic paleogeographic maps of South and Southeast Asia from early Paleocene to the Oligocene). The following time periods were designated: (1) 100–57 MYA corresponds to complete isolation of the ISC from Eurasia, (2) 57–50 MYA marks the first assumed land connections between India and modern-day Sumatra; (3) during 50–35 MYA the ISC likely continued counter-clockwise movement northwards, forming land bridges with modern-day Indo-Burma; and (4) 35–0 MYA corresponds to the firm collision and formation of a stable land connection between the ISC and Eurasia. Transition matrices between biogeographic regions for each time period are presented in Table S6.

Figure 3 Paleogeography and climate of South and Southeast Asia, 60–25 Ma.

Tectonic reconstructions modified from Hall (2012); paleoclimate reconstructions based on Morley (2018). Solid arrows indicate directions of plant dispersals (dark-green—perhumid floral elements, light-green—seasonal wet/seasonal dry elements) (from Morley, 2018); red arrows show probable areas of Microhylinae diversification and ways of their dispersal. (A) K/T boundary to early Paleocene: the isolated Indian subcontinent (ISC) is drifting northwards cradling perhumid tropical biota, Southeast Asia (SEA) has primarily seasonal wet or seasonal dry climate, no land connection between SEA and ISC, basal radiation of Microhylinae in the ISC; (B) Paleocene to early Eocene: the ISC and SEA are at the same latitude within same perhumid climate zone, first land connections between India and Sundaland via Incertus Arc, dispersal of Microhylinae from the ISC to SEA; (C) Middle Eocene: land connection between the ISC and mainland Southeast Asia (modern-day Myanmar), basal radiation of Microhyla—Glyphoglossus assemblage in SEA; (D) Oligocene: India drifts into northern high-pressure zone and seasonally dry climates predominate across the ISC and SEA, Microhyla II lineages colonize the ISC from SEA. Base Map created using https://www.simplemappr.net/.

Body size evolution

To assess body size evolution and miniaturization in Microhyla, we used weighted squared-change parsimony (Maddison, 1991) executed with Mesquite v3.31 (Maddison & Maddison, 2017). We compiled data on maximum snout-vent length (SVL) for both sexes for each Microhyla species reported in literature and/or determined from available voucher specimens (see Table S7); mensural data were taken with Mitutoyo dial caliper to the nearest 0.1 mm. SVL data for all Microhyla species are collated in Table S7.

Results

Taxa, data, and sequence characteristics

Our aligned matrix of all mtDNA data comprised 2,478 bp, included 206 samples, representing 48 species of Microhyla (96% of the currently recognized species), five species (of nine currently recognized species) of the phylogenetically closely related genus Glyphoglossus, and 24 samples from outgroup taxa (see Table S1).

The concatenated mtDNA + nuDNA dataset comprised 3,207 bp, including 118 samples from 100 ingroup and 18 outgroup taxa. Summary information on fragment lengths and variability are collated in Table S8.

Phylogenetic relationships and species groups in Microhyla

Bayesian Inference and Maximum Likelihood analyses of the mtDNA-based genealogy for Microhyla and Glyphoglossus (Figs. 4 and 5; a simplified collapsed tree is shown in Fig. S2) resulted in a topology that was generally congruent with the phylogeny obtained from the concatenated mtDNA + nuDNA data, though the latter had higher support for most nodes (Fig. 6). The BDNF gene haplotype network resulted in the species clusters which were generally congruent with the phylogenetically and morphologically recognized groups of Microhyla and were separated from each other by at least four mutational steps (Fig. S3). Most of the examined Microhyla species showed sharing of BDNF haplotypes with exception of the species pairs M. marmorata + M. pulverata, M. kodial + M. irrawaddy, and M. okinavensis + Microhyla sp. 3 (Fig. S3). Overall, since the mtDNA + nuDNA phylogenetic tree was mostly better resolved and had greater support at more nodes than the mtDNA tree, we relied on the combined mtDNA + nuDNA topology for inferring phylogenetic relationships and biogeographic history of the genus Microhyla.

Figure 4 Updated mtDNA-genealogy of the Microhyla—Glyphoglossus assemblage (full tree, part 1).

BI genealogy of Microhyla and Glyphoglossus samples examined in this study reconstructed from 2478 bp of mtDNA fragment. Values at nodes correspond to BI PP/ML BS, respectively; numbers at tips correspond to sample numbers summarized in Table S1. Colors and letters (A–I) correspond to species groups of the Microhyla—Glyphoglossus assemblage. Yellow and red bars present the results of species delimitation analyses from bGMYC and ABGD algorithms, respectively. Frog photos are given in one scale, scale bar corresponds to 10 mm, numbers near thumbnails correspond to species: (1) Microhyla nepenthicola; (2) M. borneensis; (3) M. malang; (4) M. orientalis; (5) M. mantheyi; (6) M. minuta; (7) M. achatina; (8) M. irrawaddy; (9) M. heymonsi; (10) M. pineticola; (11) M. fodiens; (12) M. fissipes; (13) M. mukhlesuri; (14) M. chakrapanii; (15) M. okinavensis; (16) M. berdmorei (Vietnam); (17) M. berdmorei (Malaysia); (18) M. picta; (19) M. pulchra; (20) M. zeylanica; (21) M. sholigari; (22) M. karunaratnei; (23) Microhyla sp. 2; (24) M. ornata; (25) M. mihintalei; (26) M. butleri; (27) M. aurantiventris; (28) M. palmipes; (29) M. annamensis; (30) M. marmorata; (31) M. pulverata; (32) M. annectens; (33) M. petrigena; (34) M. perparva; (35) M. pulchella; (36) M. arboricola; (37) Glyphoglossus molossus; (38) G. guttulatus. Photos by Nikolay A. Poyarkov, Indraneil Das, Vladislav A. Gorin, Parinya Pawangkhanant, Luan Thanh Nguyen, and Evgeniya N. Solovyeva.

Figure 5 Updated mtDNA-genealogy of the Microhyla—Glyphoglossus assemblage (full tree, part 2).

BI genealogy of Microhyla and Glyphoglossus samples examined in this study reconstructed from 2,478 bp of mtDNA fragment. Values at nodes correspond to BI PP/ML BS, respectively; numbers at tips correspond to sample numbers summarized in Table S1. For legend, see Fig. 4. Photos by Nikolay A. Poyarkov, Indraneil Das, Vladislav A. Gorin, Parinya Pawangkhanant, Luan Thanh Nguyen, and Evgeniya N. Solovyeva.

Figure 6 Maximum Likelihood tree for the “total evidence” analysis of the 3207 bp-long concatenated mtDNA + nuclear DNA dataset.

Values at nodes correspond to BEAST PP/ML BS/BI PP, respectively; black and white circles correspond to well-supported (BI PP ≥ 0.95; ML BS ≥ 90) and moderately supported (0.95 > BI PP ≥ 0.90; 90 > ML BS ≥ 75) nodes, respectively; no circles indicate unsupported nodes. Colors and letters (A–I) marking the species groups in Microhyla species complex correspond to Figs. 4 and 5, but not to Fig. 2. Photos by Nikolay A. Poyarkov, Indraneil Das, Vladislav A. Gorin, Parinya Pawangkhanant, Luan Thanh Nguyen, and Evgeniya N. Solovyeva.

The BI- and ML-analyses of mtDNA data resulted in a majority of ingroup nodes receiving high values of both PP and BS support (Figs. 4 and 5). Observed topological patterns within the Microhyla—Glyphoglossus assemblage were generally congruent across analyses and agreed well with earlier phylogenies for the group (see “Discussion”), although with generally higher node support values in our study. All analyses unambiguously supported the monophyly of the Microhyla—Glyphoglossus assemblage; however, the basal node of this radiation was not sufficiently resolved in all analyses (Fig. 6; Fig. S2). The genus Microhyla sensu lato was subdivided into two major deeply divergent groups, that we identify here as Microhyla I and Microhyla II, while monophyly of the genus with respect to Glyphoglossus was not supported according to mtDNA data (see Fig. S2); a similar pattern was also reported in the mtDNA-based genealogy of Matsui et al. (2011). Though basal divergence in the Microhyla—Glyphoglossus clade based on mtDNA + nuDNA data was also not strongly supported (Fig. 6), the topology suggesting monophyly of Microhyla I + Microhyla II agreed well with results of recent multilocus phylogenies for this group (Peloso et al., 2016; Tu et al., 2018).

All species of Microhyla I, Microhyla II, and Glyphoglossus clades were regularly grouped into one of nine well supported matrilines (Figs. 4 and 5A–5I); these same phylogenetic groupings were also revealed in the “total evidence” analysis (Fig. 6) in the BDNF haplotype network (Fig. S3) and in the most recent published phylogeny of the genus (Biju et al., 2019).

Microhyla I (subclade AI of Matsui et al., 2011) included seven major clades and 43 putative species of tiny to mid-sized terrestrial frogs (Figs. 4 and 5A–5G): (A) Clade A corresponded to M. achatina species group and received only moderate levels of monophyly support in mtDNA-genealogy (0.90/61, hereafter node support values are given for BI PP/ML BS, respectively) (Fig. 4). Genealogical relationships within this group were poorly resolved. Phylogenetic positions of M. fodiens from central Myanmar, the M. heymonsi complex, and M. pineticola from Indochina were unresolved. Other species form a strongly supported monophyly (1.0/98), which is further subdivided into two subclades: (A1) comprising species from Sundaland (M. borneensis, M. nepenthicola, Microhyla sp. 1 from Sabah, M. malang, M. orientalis, M. mantheyi) and southern Vietnam (M. minuta) (0.97/68), and (A2) with species from Sundaland (M. achatina, M. gadjahmadai), Myanmar (M. irrawaddy, Microhyla sp. 4 from northern Myanmar), and southern India (M. kodial) (1.0/87). Monophyly of clade A was not supported in the “total evidence” analysis (0.72/50; Fig. 6), while the clade including all members of M. achatina species group except M. fodiens received moderate support (0.91/62; Fig. 6). The M. achatina species group occupied a more central position in the BDNF gene haplotype network (Fig. S3) and was generally poorly delineated, in agreement with results of Garg et al. (2019).

(B) Clade B corresponded to M. fissipes species group (1.0/100) and consisted of two well-supported subclades (Fig. 4): (B1) included species from Indochina and southern mainland China and Taiwan (M. fissipes, M. mukhlesuri), and species from eastern India, Bangladesh and the Andaman Islands (M. mymensinghensis, M. chakrapanii) (1.0/100); (B2) encompassing species from mainland China (M. mixtura, M. beilunensis, M. fanjingshanensis) and the Ryukyus (M. okinavensis, Microhyla sp. 3 from Yaeyama Archipelago) (1.0/98). In the BDNF gene haplotype network, M. okinavensis and Microhyla sp. 3 (B2) were distantly placed from members of the M. fissipes species group with a minimum of 14 mutational steps (Fig. S3), also agreeing well with results of Garg et al. (2019). The “total evidence” analysis suggested sister group relationships between M. fanjingshanensis and M. okinavensis (0.96/55; Fig. 6).

(C) Clade C included the M. berdmorei complex and M. pulchra from Indochina and southern China, as well as M. picta from southernmost Vietnam (1.0/100). Clade C was recovered as sister clade to a clade A + B (1.0/90 for mtDNA, and 1.0/96 for “total evidence” datasets, respectively; see Figs. 5 and 6); a similar topology of phylogenetic relationships was also reported by Biju et al. (2019). In our BDNF gene haplotype network, M. berdmorei species complex is separated from M. pulchra + M. picta by at least eight mutational steps (Fig. S3).

(D) Clade D encompassed species from Sri Lanka (M. zeylanica, M. karunaratnei) and southern India (M. laterite, M. sholigari, M. darreli) (D1, see Fig. 5), but also included species from northeastern India (M. eos), western Thailand (Microhyla sp. 2 from Tenasserim) and Thai-Malay Peninsula (M. superciliaris). The BDNF haplotype network suggests distant placement of Microhyla sp. 2, separated by at least seven mutational steps from other members of the M. superciliaris group (Fig. S3). The “total evidence” analysis strongly suggested sister group relationships between clades D and F (0.96/67; Fig. 6), in agreement with the phylogeny presented by Biju et al. (2019).

(E) Clade E included species with distribution on the Indian Subcontinent and Sri Lanka, including M. ornata complex (E1, M. ornata, M. nilphamariensis, M. taraiensis) and M. rubra complex members (E2, M. rubra, M. mihintalei). In the BDNF gene haplotype networks, subclades E1 and E2 are separated from each other by at least three mutational steps (Fig. S3). MtDNA data suggested a tendency for joining the Clades D and E in a monophylum (0.91/75; Fig. 5), but it was not supported by the “total evidence” analysis, which instead placed Clade E as a sister group to the clade A + B + C, though with no support (0.70/37; Fig. 6). A similar topology was also proposed by Biju et al. (2019) but they also had insignificant node support (0.67/80). The monophyly of the clade joining matrilines A–E was moderately supported by mtDNA (Fig. 5), but not by the mtDNA+nuDNA dataset (Fig. 6).

(F) Clade F corresponds to M. butleri species group and joined the M. butleri complex from southern China and Southeast Asia with M. aurantiventris from central Vietnam (1.0/100). Clade F was strongly supported as a sister lineage with respect to Clade D based on the “total evidence” analysis (0.96/67) (Fig. 6) and the BDNF gene haplotype network (Fig. S3, separated by at least eight mutational steps). Monophyly of the clade joining D + E had moderate support in earlier phylogenies of the genus (0.91/81, see Biju et al., 2019).

(G) Clade G was represented by the M. palmipes species complex from Java and Sumatra; its phylogenetic position was poorly supported. In contrast to earlier data, suggesting sister relationships between M. palmipes and the group joining A + B + C + E clades (0.99/56; Biju et al., 2019), our “total evidence” analysis suggests sister group relationships of M. palmipes with respect to clade D + F but with moderate support (0.92/41; Fig. 6). The BDNF gene haplotype network also places M. palmipes closer to Clade D, separated by at least seven mutational steps (Fig. S3).

Microhyla II (subclade AIII of Matsui et al. (2011)) was represented by a single clade (H) and included nine nominal species of tiny to small-sized terrestrial or semi-arboreal frogs (Fig. 5H):(H) Clade H corresponds to M. annectens species group and was further subdivided in two subclades: (H1) included species from Borneo (M. perparva, M. petrigena), the Thai-Malay Peninsula (M. annectens), and Annamite Mountains of Indochina (M. annamensis, M. marmorata and M. pulverata) (1.0/88); and (H2) joined species from central (M. nanapollexa) and southern parts of Annamite Mountains (M. arboricola, M. pulchella). M. marmorata was recovered paraphyletic, with respect to M. pulverata (1.0/100; Fig. 5). In the BDNF gene haplotype network, Clade H is separated from Microhyla I by at least 11 mutation steps; while H1 and H2 subclades are poorly delineated (Fig. S3).

Finally, the large-sized fossorial frogs of the genus Glyphoglossus (subclade AII of Matsui et al. (2011)) was represented in our analysis by five of the nine recognized species; G. capsus from Borneo was recovered as a sister group to the clade that joined all other species from Indochina and Malay Peninsula (Fig. 5) (0.98/74). The most recent phylogenies of Microhyla did not include any species of Glyphoglossus (Garg et al., 2019; Biju et al., 2019). In our work, the genus Glyphoglossus is separated from Microhyla by at least eight mutational steps in the BDNF gene haplotype network (Fig. S3).

Species delimitation analyses

The BI matrilineal genealogy for the 48 nominal Microhyla species provided an initial assessment of species-level relationships (Figs. 4 and 5). Uncorrected genetic p-distances in 16S rRNA mtDNA gene within and among Microhyla species are given in Table S9. In addition to currently recognized taxa, our genealogy also depicted at least four lineages of Microhyla that likely represent unrecognized species (Microhyla spp. 1–4), and a number of deep lineages within species complexes (intraspecific genetic differences p > 1.5%, for example, M. malang, M. achatina, M. gadjahmadai, M. heymonsi, M. okinavensis, M. berdmorei, M. butleri, M. palmipes, M. petrigena, M. perparva and M. arboricola; see Table S9), potentially constituting cryptic species diversity. Generally, interspecific divergences were p > 3.0%, but in some cases, intraspecific divergences exceeded interspecific divergences (Table S9).

To assess the number of putative species-level lineages within the genus Microhyla, we implemented two alternative approaches to species delimitation via tree-based bGMYC and distance-based ABGD analyses. These methods show varying performance depending on sample and population sizes, speciation rates, and other parameters, with bGMYC showing a tendency to oversplit, while ABGD often overlumps putative species; however, when these methods agree, the resulting delimitation gains plausibility (Dellicour & Flot, 2018). The combined results of both bGMYC and ABGD analyses (Figs. 4 and 5; summarized in Table S10) resolved all described species of Microhyla, except for M. pulverata, which was indistinguishable from M. marmorata. This result is also corroborated by the mtDNA-genealogy (Fig. 5), the BDNF gene haplotype network (Fig. S3), and divergence data for 16S rRNA gene (Table S9). Both analyses suggest that species diversity within Microhyla is greatly underestimated: for 48 recognized Microhyla species included in our genealogy, the bGMYC analysis recovered 81, and ABGD recovered 63 species-level lineages (Table S10) (note that the two species missing from our analysis, M. fusca and M. darevskii, are not included in these totals). In most cases, analyses had congruent results, however in 14 species, bGMYC proposed more groups than were recovered by ABGD. Nonetheless, species delimitation analyses strongly indicate the presence of many unrecognized species-level lineages by further partitioning M. heymonsi (into 7–8 species), M. bermodrei (3–4 species), M. malang (3 species), M. butleri (2–4 species), and M. gadjahmadai, M. okinavensis, M. palmipes, M. perparva, M. petrigena, M. achatina and M. arboricola (each with 2 species) (Figs. 4 and 5; Table S10). For nine other taxa, the analyses gave incongruent results, with bGMYC splitting and ABGD lumping these disparate taxa (M. beilunensis, M. chakrapanii, M. mantheyi, M. mixtura, M. mukhlesuri, M. orientalis, M. superciliaris, Microhyla sp. 3 and Microhyla sp. 4) (Table S10).

Divergence times estimation

The resulting BEAST chronogram (see Fig. S4; BEAST results for the ingroup are further detailed in Fig. 2) elucidates that the most recent common ancestor (MRCA) of Microhyla and Glyphoglossus originated between late Paleocene and early Eocene, ca. 50.8 Ma (44.1–57.0), and agrees with the analysis of Feng et al. (2017), ca. 48.8 Ma (45.9–53.2), and is notably earlier than the estimate by Garg & Biju (2019) as 61.5 Ma (56.6–66.5). The group Microhyla + Glyphoglossus radiated within a relatively narrow time period in the middle Eocene ca. 43.8 Ma (38.7–49.1), slightly younger than estimates of Garg & Biju (2019), who estimated this cladogenetic event at ca. 48.7 Ma (44.1–53.2). Diversification within the genus-level endemic radiations of Microhyla I, Microhyla II, and Glyphoglossus clades started in early to mid-Oligocene (from 35 to 25 Ma), generally agreeing with Garg & Biju (2019). Estimated node-ages and the 95% highest posterior density (95% HPD) for the main nodes are summarized in detail in Table S11.

Historical biogeography

We were able to reconstruct biogeographic processes (vicariance, dispersal, and colonization routes) and ancestral areas (Fig. 2) for the genus Microhyla using the RASP biogeographic analysis. According to both the DEC and the S-DIVA model, the MRCA of Microhyla + Glyphoglossus (node 12, Fig. 2) most likely inhabited Eastern Indochina. Eastern Indochina was also reconstructed as an ancestral range for Microhyla II and Glyphoglossus lineages (nodes 18 and 12, respectively, Fig. 2), and Microhyla I likely originated in western Indochina (node 30, Fig. 2). Microhyla II expanded its range to Borneo and the Malay Peninsula, but Microhyla I dispersed more widely to all biogeographic regions within the modern range of the genus, including at least five independent dispersal events from western Indochina to the Indian Subcontinent (Fig. 2). Results of our analyses suggest numerous cases of dispersal from the Asian mainland to islands including Sundaland, but only a single case of reverse dispersal (from Borneo to the Malay Peninsula, see Fig. 2).

Body size evolution modeling

Reconstructions of maximum SVL ancestral states and their evolution in Microhyla and Glyphoglossus are shown in Fig. 7. Maximum adult SVL differed substantially for males and females, so data were analyzed separately for each sex. Glyphoglossus (adult male SVL 30.0–95.0 mm) are generally larger compared to Microhyla (adult male SVL 10.6–35.0 mm); the MRCA of the Microhyla + Glyphoglossus assemblage is reconstructed as a mid-sized frog (25.0–30.0 mm SVL for males, 30.0–35.0 mm for females). Most species of Microhyla were found to be smaller (male SVL roughly 11.5–25.0 mm) than their common ancestor, with five cases of miniaturization (male SVL 11.5–15.0 mm). However, both Glyphoglossus and M. berdmorei species groups of Microhyla I have subsequently increased their body size (up to 105 mm in Glyphoglossus molossus, up to 35 mm in M. berdmorei) (Fig. 7).

Figure 7 Body size evolution among members of the Microhyla—Glyphoglossus assemblage.

See Table S7 for SVL data. Color of branches corresponds to average SVL in males (A) and females (B) (in mm).

Discussion

Updated phylogenetic relationships of Microhyla

In phylogenetic systematics, extensive taxon sampling increases accuracy and support of evolutionary relationships (Zwickl & Hillis, 2002). Herein, we present an updated phylogenetic study of the genus Microhyla, with the most complete taxon sampling including 48 of the 50 currently recognized species. The absent taxa in our study are M. fusca and M. darevskii—two enigmatic species from central and southern Vietnam. Microhyla fusca was described from a single specimen collected from southern Vietnam (Andersson, 1942); no additional specimens of this species were reported after its discovery despite numerous field survey efforts. Microhyla darevskii was described from a series of formalin-fixed specimens and morphologically resembles members of M. berdmorei species complex (Poyarkov et al., 2014); our repeated attempts to get DNA data from the type series of M. darevskii were not successful. Further studies of museum specimens and increased field survey efforts are required to clarify the taxonomic status and phylogenetic affinities of M. fusca and M. darevskii. We also did not sample Microhyla maculifera, a small-sized species described from Danum Valley in Sabah, Borneo (Inger, 1989); it might correspond to Microhyla sp. 1 in our analysis, collected from its type locality. Unfortunately, specimens of Microhyla sp. 1 included in our phylogenetic analyses were not available for morphological examination; we thus hesitate to identify this population as M. maculifera pending further morphological study.

The first phylogenetic study of Asian microhylids by Matsui et al. (2011) demonstrated paraphyly of Microhyla with respect to Glyphoglossus (at that time including Calluella). Subsequent multilocus phylogenetic studies (Peloso et al., 2016; Tu et al., 2018; Garg & Biju, 2019) strongly suggested sister group relationships between Glyphoglossus and Microhyla sensu lato, but still recognized the presence of two deeply divergent lineages within Microhyla. Our time-tree suggests that divergence between the two major clades of Microhyla I and Microhyla II happened soon after the basal split within the Microhyla + Glyphoglossus assemblage during the middle Eocene and similar divergence time estimates were obtained by a recent analysis by Garg & Biju (2019). Robust phylogenies coupled with examination of external morphological and osteological characters are required to assess the evolutionary differences among the three subclades of the Microhyla + Glyphoglossus assemblage, likely warranting recognition as three distinct genera.

The first classifications of the genus Microhyla into species groups were based exclusively on morphological characters (Parker, 1934; Dubois, 1987; Fei et al., 2005) but have not been supported by more recent molecular data (Matsui et al., 2011; Garg et al., 2019). Matsui et al. (2011) assessed genealogical relationships among 20 Microhyla species and proposed recognition of five distinct species groups within the genus. Recently Garg et al. (2019) proposed a new scheme for grouping Microhyla species and recognized six species groups based on more extensive sampling of 33 species. Our phylogenetic hypothesis mostly agrees with these earlier proposed phylogenies of the genus (Matsui et al., 2005, 2011; Matsui, Hamidy & Eto, 2013; Matsui, 2011; Hasan et al., 2012, 2014a; Howlader et al., 2015; Peloso et al., 2016; Wijayathilaka et al., 2016; Seshadri et al., 2016; Yuan et al., 2016; Khatiwada et al., 2017; Tu et al., 2018; Garg et al., 2019; Nguyen et al., 2019; Poyarkov et al., 2019; Biju et al., 2019), and a much more extensive taxon sampling allows us to revise species groups more accurately in Microhyla (Fig. 6).

The Microhyla achatina group (Clade A, Figs. 4 and 6; part of group AId2 of Matsui et al. (2011)) includes species mostly from Southeast Asia (M. achatina, M. gadjahmadai, M. heymonsi, M. pineticola, M. minuta, M. mantheyi, M. orientalis, M. malang, M. nepenthicola, M. borneensis, Microhyla sp. 1), but also from Myanmar (M. irrawaddy, M. fodiens, Microhyla sp. 4) and southern India (M. kodial). This group includes tiny (male SVL 10.6 mm, Das & Haas, 2010) to mid-sized (male SVL 29.1 mm, Poyarkov et al., 2019) frogs with lateral nostrils; dorsal skin shagreened to prominently granular skin; small disks on digits usually present and typically bearing terminal grooves; toe webbing rudimentary or absent; inner metatarsal tubercle rounded or oval-shaped, outer metatarsal tubercle rounded and small, or large, shovel-shaped (in M. fodiens, Poyarkov et al., 2019); usually with a mid-dorsal line or skinfold and a light streak extending from posterior corner of eye to axilla (Garg et al., 2019). The phylogenetic position of M. minuta, M. pineticola, and M. borneensis sensu stricto was assessed for the first time. Vietnamese M. minuta was recovered as a member of subgroup A1, joining species from Peninsular Malaysia (M. mantheyi), Java + Bali (M. orientalis), and Borneo (M. malang, M. nepenthicola, M. borneensis, Microhyla sp. 1). Our study also supports previously reported placement of south Indian M. kodial in one clade with Myanmar species M. irrawaddy and M. sp. 4 (Poyarkov et al., 2019); this clade is sister to the M. achatina + M. gadjahmadai clade from Java and Sumatra (subgroup A2). Unfortunately, phylogenetic positions of the M. heymonsi complex and M. pineticola remain unresolved. The morphologically different semi-fossorial M. fodiens from central Myanmar, previously identified as M. rubra (Wogan et al., 2008; Peloso et al., 2016), or M. cf. berdmorei (Garg et al., 2019), clearly belongs to this group and forms a highly divergent lineage (Poyarkov et al., 2019).

The Microhyla fissipes group (Clade B, Figs. 4 and 6; part of group AId2 of Matsui et al. (2011)) joins species from East Asia (M. fissipes, M. mixtura, M. beilunensis, M. okinavensis, M. fanjingshanensis and Microhyla sp. 3), Indochina, and eastern India (M. mukhlesuri, M. mymensinghensis, M. chakrapanii). We also agree with Garg et al. (2019) in recognizing this species group as distinct from the M. achatina group, though monophyly of the latter is only moderately supported (Fig. 4). Members of this group are in general morphologically similar to M. achatina species group, but can be distinguished from the latter by finger tips lacking disks, toe tips rounded or bearing tiny disks lacking terminal grooves; by inner metatarsal tubercle elongated, outer metatarsal tubercle small, rounded; and by a dark band running from canthus rostralis posteriorly towards groin and posterior parts of belly (Garg et al., 2019). Our phylogeny recovered two well-supported subgroups within the M. fissipes group corresponding to Indochinese (B1) and East Asian (B2) taxa (Fig. 6) (Yuan et al., 2016), supported sister relationships of M. chakrapanii and M. mymensinghensis (Garg et al., 2019), and suggested sister group relationships of M. fanjingshanensis and M. okinavensis (Fig. 6).

The Microhyla berdmorei group (Clade C, Figs. 5 and 6; group AId1 of Matsui et al. (2011)) encompasses the largest Microhyla species and, according to our data, includes the wide-ranging M. berdmorei complex (SVL up to 45.6 mm; Southeast Asia), M. pulchra (SVL up to 36.5 mm; Indochina and southern China), and stout-bodied semi-fossorial M. picta from southern Vietnam (SVL up to 33.4 mm). The phylogenetic position of M. picta was investigated here for the first time, and was recovered as sister taxon to M. pulchra. Members of the M. berdmorei species group exhibit considerable morphological differentiation: smooth to sparsely granular skin on dorsum; finger tips rounded, toe tips rounded or bearing tiny disks with or without terminal grooves; toe webbing from rudimentary to complete reaching toe disks; inner metatarsal tubercle oval, outer metatarsal tubercle from small to large, shovel-shaped (in M. picta); lacking mid-dorsal line or skinfold; with a light streak extending from posterior corner of eye to axilla; and a characteristic bright-yellow coloration of the groin and posterior parts of the belly. Based only on morphological characteristics, the unsampled M. darevskii (absent in our phylogeny), likely belongs to this species group.

The Microhyla superciliaris group (Clade D; part of group AIc of Matsui et al. (2011)) joins small-sized species from South and Southeast Asia, and includes a well-supported clade of southern Indian and Sri Lankan species (Figs. 5 and 6, D1, M. zeylanica, M. laterite, M. sholigari, M. karunaratnei and M. darreli; M. zeylanica group of Garg et al. (2019)), three species from northeastern India (M. eos), western Thailand (Microhyla sp. 2) and the Malay Peninsula (M. superciliaris). Morphologically members of this group are small-sized (male SVL under 21.5 mm), have dorsal orientation of nostrils; smooth to granular skin on dorsum; finger disks rounded or with weak disks lacking terminal grooves; toe disks having terminal grooves; inner metatarsal tubercle oval-shaped, outer metatarsal tubercle small and rounded; toe webbing reduced or well-developed; mid-dorsal skinfold generally present; with a light streak from posterior eye corner to axilla; and often with contrasting black and white blotches on belly. Garg et al. (2019) proposed recognizing M. zeylanica species group for taxa inhabiting peninsular India + Sri Lanka, but Biju et al. (2019) reported M. eos from northeastern India as a sister lineage of this clade, refraining from assigning this species to any species group. Herein, we propose recognizing the M. superciliaris species group for taxa inhabiting Southeast Asia, northeastern and southern India, and Sri Lanka.

The Microhyla ornata group (Clade E; group AIb of Matsui et al. (2011)) is comprised of species exclusively occurring in the Indian Subcontinent, subdivided into two groups, E1 (M. ornata, M. nilphamariensis, and M. taraiensis; corresponds to M. ornata group of Garg et al. (2019)), and E2 (M. rubra, M. mihintalei; corresponds to M. rubra group of Garg et al. (2019)) (Figs. 5 and 6). Morphologically M. ornata group includes small to mid-sized species with lateral nostrils; shagreened to granular dorsal skin; tips of digits lacking disks and terminal grooves; toe webbing rudimentary; inner and outer metatarsal tubercles present, latter may be enlarged; middorsal line or skinfold present; body flanks with dark band from nostrils to groin; and a light streak from posterior eye corner to axilla. Though Garg et al. (2019) proposed recognizing the more robust, semi-fossorial species M. rubra and M. mihintalei as a distinct M. rubra species group, we do not follow their scheme since phylogenetic relationships within Clade E are well resolved and subclades E1 and E2 are closely related (Matsui et al., 2011). In addition, adaptations to burrowing lifestyle are not unique for M. rubra but are found in other lineages of Microhyla as well, such as M. fodiens of Clade A, and M. picta of Clade C.

The Microhyla butleri group (Clade F; part of group AIc of Matsui et al. (2011); see Figs. 5 and 6) includes the M. butleri complex from Southeast Asia and southern China and the closely related M. aurantiventris from central Vietnam. Morphologically members of this group show a dorsolateral nostril position; prominently granular dorsal skin; presence of weak disks on digits bearing terminal grooves; moderately developed webbing on toes; inner and outer metatarsal tubercles small; middorsal line or skinfold present; characteristic “teddy-bear”-shaped dark marking on dorsum edged with light color; body flanks lacking dark band from nostrils to groin; and a light eye-axilla streak present (Nguyen et al., 2019).

The Microhyla palmipes group (Clade G; group AIa of Matsui et al. (2011)) presently includes a single species, M. palmipes, from Java, Sumatra, and adjacent offshore islands in Indonesia, and according to our data, likely represents a species complex (see below). Morphological data on M. palmipes are scarce; they are small-sized frogs (male SVL 16.0 mm) with lateral nostrils; shagreened skin on dorsum; weak disks on digits lacking terminal grooves; moderately developed webbing on toes; inner and outer metatarsal tubercles small; middorsal line absent; dark markings on flanks and a light eye-axilla streak present (Bain & Nguyen, 2004; Poyarkov et al., 2014).

Finally, the M. annectens group (Clade H of Microhyla II; group AIII of Matsui et al. (2011); Figs. 5 and 6) encompasses tiny (male SVL 13.2 mm) to mid-sized (male SVL to 21.6 mm) frogs from Southeast Asia, and according to our phylogeny, is subdivided into two subgroups: H1 comprising species from Malayan Peninsula (M. annectens), Annamite Mountains in Vietnam (M. annamensis, M. marmorata, and M. pulverata), and Borneo (M. petrigena and M. perparva); and H2 including species from mountains of central and southern Vietnam (M. arboricola, M. pulchella, M. nanapollexa). M. marmorata was found to be paraphyletic with respect to M. pulverata. Morphologically, M. annectens group members are characterized by a relatively short body; lateral position of nostrils; sparsely granular to tubercular dorsal skin; complete toe webbing with well-developed, flattened, and slightly expanded disks on digits bearing terminal grooves; inner metatarsal tubercle present, and outer metatarsal tubercle present or absent. Further morphological studies are required to examine morphological differentiation among M. annectens group members. Phylogenetic positions of M. annamensis, M. marmorata, M. pulverata, M. arboricola, and M. pulchella are for the first time reported in the present study.

Our updated phylogeny reveals several lineages of Microhyla that likely represent undescribed species: Microhyla sp. 1 from Sabah, Malaysia (corresponds to Microhyla sp. 1 of Matsui et al. (2011)), Microhyla sp. 2 from western Thailand (previously not reported), Microhyla sp. 3 from Yaeyama Archipelago (previously referred to as M. okinavensis), and Microhyla sp. 4 from northern Myanmar (reported as Microhyla sp. A by Mulcahy et al. (2018)). Our study also recovered significant diversity within wide-ranging species complexes that might comprise undescribed cryptic species (N = 31). This suggests that the taxonomy of the genus Microhyla still remains largely incomplete.

Indian Collision and historical biogeography of Microhyla

The origin of Asian microhylids, including the subfamily Microhylinae, is connected with a break-up of Gondwana and the Indian Collision (Van Bocxlaer et al., 2006; Van der Meijden et al., 2007). Most likely, ancestors of Asian Microhylidae subfamilies diverged and diversified on the Indian Plate during its long isolation and northward drifting in the late Cretaceous and Paleocene (Bossuyt & Milinkovitch, 2001; Kurabayashi et al., 2011; De Sá et al., 2012) (see Fig. 3A). The basal divergence of the subfamily Microhylinae most likely took place on the Indian Plate prior to its first contact with Eurasia and the ISC is regarded as the original source of Microhylinae diversity (Garg & Biju, 2019). However, Southeast Asia (not the ISC) presently harbors the largest number of Microhylinae lineages and species (Frost, 2020).

Several recent biogeographic studies suggested that collision of the ISC with the Asian mainland was a more complicated process than previously conceived, implicating early opportunities for faunal exchange between the ISC and present-day Southeast Asia (Klaus et al., 2010; Li et al., 2013; Grismer et al., 2016). The “Out of India” scenario, with early dispersal from the ISC to Sundaland via brief land connection in early Eocene, has also been proposed for the Microhylinae (Garg & Biju, 2019). Recent progress in tectonic plate modeling further corroborates the possibility for biotic exchange between the ISC and Sundaland via the Incertus Arc land bridge, starting 55–50 Ma (Fig. 3B), although exact timing and configuration of the landmasses remains under debate (Hall, 2012; Ding et al., 2017). Interestingly, paleoclimate reconstructions suggest that modern-day megathermal angiosperm-dominated tropical forests also originated in the ISC. They later dispersed from there and became established across Sundaland starting about 50 Ma (Morley, 2018) coinciding with the onset of a perhumid climate in Southeast Asia (Fig. 3B).

The present time tree analysis indicates that the ancestral radiation of Microhyla + Glyphoglossus into three main lineages (Microhyla I, Microhyla II and Glyphoglossus) happened during a short time frame in the middle Eocene (ca. 43.8 Ma), slightly later than previous estimates (48.7 Ma; Garg & Biju, 2019). Our biogeographic analysis strongly suggests that the Microhyla + Glyphoglossus MRCA, as well as the Microhyla I + Microhyla II ancestor, inhabited Eastern Indochina (Fig. 2), which was connected to Sundaland (Fig. 3C). Thus, our results support the Southeast Asian origin of the Microhyla—Glyphoglossus assemblage in contrast to the hypothesis by Garg & Biju (2019), that suggested the dispersal of Microhyla from the ISC into Asia from the Oligocene to the Miocene.

The Microhyla II clade remained largely within its ancestral area with most members of the group inhabiting Eastern Indochina and a few species dispersing to Borneo and the Malay Peninsula (Fig. 2). Compared to other Microhyla species groups, members of the Microhyla II clade are generally small and associated with perhumid montane evergreen forests or tropical rainforests; they do not occur in lowland seasonally dry areas. In fact, in Indochina, their distribution is restricted to mountainous areas (Parker, 1934; Poyarkov et al., 2014).

On the contrary, members of the Microhyla I clade dispersed widely and achieved a pan-Oriental distribution (see Figs. 1 and 2). Members of this clade are diverse ecologically and morphologically. They vary in body size from the smallest to the largest Microhyla taxa, occupy habitats that include open seasonally dry savannahs (Parker, 1934), and several species within the Microhyla I clade evolved adaptations towards digging and estivation (M. rubra, M. mihintalei, M. picta, and M. fodiens; see Poyarkov et al., 2019). The MRCA of Microhyla I is hypothesized to inhabit western Indochina, the same region reconstructed as ancestral for all major internal nodes within Microhyla I, and for a number of included lineages (M. heymonsi, M. fissipes, M berdmorei, and M. superciliaris species groups), respectively (Fig. 2).

Drifting of the ISC northwards led to a collision of the Indian plate with Eurasia from the Oligocene to the Miocene (Aitchison & Ali, 2012; Hall, 2012) (Fig. 3D). At the same time, the uplift of the Himalayas, coinciding with the middle Miocene thermal maximum, initiated the subsequent Miocene strengthening of the Indian monsoon and entailed the expansion of seasonally dry conditions across the northern parts of the Indian peninsula and Indochina. This resulted in the disappearance of closed tropical forests over much of the ISC (Morley, 2018) (Fig. 3D). Starting in the Oligocene, Indochina became the source of evergreen and seasonally dry floral elements that dispersed to the ISC with ongoing climate aridification (Morley, 2000). These conditions presumably facilitated colonization of the Indian Subcontinent by Microhyla I lineages.

Our biogeographic analysis reveals at least five independent cases of Microhyla I dispersal from Western Indochina to the ISC (see Fig. 2). Two of these took place in Late Oligocene–Early Miocene: the M. superciliaris species group (29.2–22.5 Ma, Fig. 2, 1) and M. ornata species group (31.4–18.8 Ma, Fig. 2, 2). Both lineages underwent significant diversification in the ISC and reached as far south as Sri Lanka. Three other cases of the ISC colonization by Microhyla I include more recent dispersal events in Late Miocene—Pliocene by M. berdmorei (6.9–4.1 Ma, Fig. 2, 3) and M. fissipes species groups (M. mymensinghensis, 7.2–5.1 Ma, Fig. 2, 4) to northeastern India and Bangladesh, with the only case of dispersal to southern peninsular India being the species of M. achatina group in the Middle Miocene (M. kodial, 16.2–8.9 Ma, Fig. 2, 5). The confusing phylogenetic position of M. kodial within the Southeast Asian M. achatina species group originally inspired the hypothesis that this might be a result of a human-mediated dispersal and introduction (Vineeth et al., 2018). However, subsequent discovery of its sister species M. irrawaddy and Microhyla sp. 4 in central Myanmar have made the hypothesis of natural dispersal of the M. achatina species group members from Southeast Asia to the ISC more plausible. Interestingly, M. irrawaddy inhabits seasonally dry savannah areas, with minimal rainfall (Poyarkov et al., 2019). Hence, progressing aridification of the northern and central parts of the ISC starting in the late Miocene (Deepak & Karanth, 2018) could have created suitable habitats facilitating dispersal of M. kodial ancestors. Generally, western Indochina played an important role for the Microhyla I clade (Fig. 2) as this territory likely represents a “stepping stone” area connecting Southeast Asia and the ISC (Fig. 3D).

Diversification within Microhyla species group-level endemic radiations started in the Late Oligocene–Early Miocene and initiated multiple dispersals from Asian mainland to present-day islands and archipelagos (Fig. 2). These include multiple colonizations of Sundaland from both Indochina and the Malay Peninsula (by M. annectens, M. palmipes, M. berdmorei and M. achatina species groups; Fig. 2). This is not surprising, since these territories are believed to have been a single landmass throughout most of Cenozoic (Cannon, Morley & Bush, 2009; Woodruff, 2010; Hall, 2012). Microhyla superciliaris and M. ornata species groups experienced at least four independent dispersal events from southern India to Sri Lanka, corroborating results of recent studies, and suggesting a complex history of dispersals between these regions (Harikrishnan et al., 2012; Pyron et al., 2013; Agarwal et al., 2017; Karunarathna et al., 2019). Our study confirms placement of M. chakrapanii from Andaman Islands inside the Southeast and East Asian M. fissipes species group as sister species of M. mymensinghensis (see Garg et al., 2019). This confirms the faunal similarity of the Andamans with Southeast Asia rather than with peninsular India (see Das, 1994, 1999). Finally, members of M. achatina, M. butleri, and M. fissipes species groups have dispersed several times from the Asian mainland to East Asian islands: Taiwan (M. fissipes, M. heymonsi, and M. butleri), and two times independently colonized the Ryukyus (M. okinavensis and Microhyla sp. 3). These results also corroborate data that suggested faunal exchanges between Eurasian continent and East Asian islands (Ota, 1998; You, Poyarkov & Lin, 2015; Yuan et al., 2016; Wang et al., 2017; Nguyen et al., 2020) and require further study (Lee et al., 2016; Tominaga et al., 2019).

Interestingly, our phylogeny suggests numerous dispersal events from the Asian mainland to islands, with almost no dispersals back to the mainland (except for the M. annectens ancestor that is hypothesized to have dispersed from Borneo to Malay Peninsula, see Fig. 2). This also supports results of De Bruyn et al. (2014) who demonstrated that colonization events from younger Asian islands are comparatively rare, rather showing increased levels of immigration events as compared to Indochina or Borneo. Further studies are required to elucidate the role of islands in producing and preserving diversity in Microhyla frogs.

Implications for body size evolution in Microhyla

Miniaturization is a widespread and interesting morphological and ecological phenomenon in amphibians (Hanken, 1985; Hanken & Wake, 1993; Rieppel, 1996). It is common in several groups of frogs (Clarke, 1996; Lehr & Coloma, 2008) and reaches extremes in the Microhylidae (Kraus, 2011; Rittmeyer et al., 2012; Oliver et al., 2017; Rakotoarison et al., 2017; Scherz et al., 2019). The smallest Microhylinae and the smallest terrestrial vertebrate in Asia belong to Microhyla and include representatives of two different lineages within the genus: Microhyla I (M. nepenthicola, adult male size from 10.6 mm; see Das & Haas, 2010) and Microhyla II (M. perparva, males 10.5–11.9 mm, and M. arboricola, adult male size from 13.2 mm; see Inger & Frogner, 1979; Poyarkov et al., 2014). Osteological consequences of miniaturization in Microhyla are not well studied yet diminutive species in Microhyla II clade show partial (M. arboricola) or almost complete (M. perparva, M. nanapollexa) reduction of the first finger. Similar patterns have also been reported in other miniature microhylids (Kraus, 2011; Rakotoarison et al., 2017), however, patterns and drivers of body size evolution in Microhyla remain poorly understood.

In Microhyla, males generally tend to be smaller than females (see Table S7), yet our analyses revealed generally similar patterns of body size evolution in both sexes (Fig. 7). According to the most plausible scenario, the common ancestor of the Microhyla + Glyphoglossus assemblage was a mid-sized frog (male SVL 25–30 mm, female SVL 30–35 mm). Body size increased in the Glyphoglossus clade (up to 105 mm SVL), was slightly reduced in the Microhyla I clade, and significantly reduced in the Microhyla II clade ancestors (see Fig. 7). Microhyla II members are all small-sized (<25 mm SVL in males) and three species of this group reach extreme miniaturization (<15 mm SVL in males; M. arboricola, M. perparva, M. nanapollexa). Microhyla I clade shows significant variation in body size: the M. berdmorei species group and, to a lesser extent, several species in other lineages (including M. fodiens, M. rubra, M. mihintalei, M. aurantiventris) demonstrate increased body sizes, while members of two lineages within M. achatina (M. nepenthicola + M. borneensis + Microhyla sp. 1) and M. superciliaris species groups (M. superciliaris + Microhyla sp. 2) are diminutive (Fig. 7).

Our analyses suggest that adult body size has independently changed several times in the evolution of the Microhyla + Glyphoglossus assemblage. At least two lineages show an increase in body size, while four other lineages demonstrate extreme miniaturization (Fig. 7). Significant increases in body size in these frogs seem to be connected with a fossorial life style. For example, all members of Glyphoglossus as well as some large Microhyla (M. picta, M. fodiens, M. rubra and M. mihintalei) have stout body habitus and are excellent burrowers (Poyarkov et al., 2019). Most of these species inhabit open seasonally dry habitats at low elevations and burrowing is an important strategy for estivation during dry periods. Increased body size might not only facilitate digging, but is also advantageous due to lower surface-area to volume ratios, hence leading to less evaporative water loss during estivation (Tracy, Christian & Tracy, 2010).

Evolutionary causes for extreme miniaturization in frogs remain under debate (Scherz et al., 2019), but are probably connected with life history strategies, such as exploiting new food resources (Lehr & Coloma, 2008) or adaptation to specific microhabitats such as leaf-litter or moss (Kraus, 2011). Scherz et al. (2019) noted that one miniaturized Malagasy microhylid species is arboreal and breeds in water-filled leaf-axil phytotelmata, while all others are terrestrial. Interestingly, at least the three smallest known Microhyla species are also obligatory phytotelm-breeders and reproduce in water-filled pitcher-plants (M. nepenthicola, see Das & Haas, 2010; and M. borneensis, see Parker, 1928) or water-filled tree hollows (M. arboricola, see Poyarkov et al., 2014; Vassilieva et al., 2017). Among all other Microhyla species, M. arboricola is the only known arboreal species with a unique reproductive mode: developing tadpoles of this species are obligately oophagous (Vassilieva et al., 2017). Microhyla arboricola has a reduced clutch size (16 ± 8 eggs), compared to other Microhyla species (usually over 400 eggs per clutch) (Vassilieva et al., 2017). Egg size appears to be one of the main constraints for miniaturization in animals (Polilov, 2015). On one hand, reduced clutch size might favor the choice of phyototelmata for reproduction due to the absence or low density of predators in such habitats; on the other hand, diminutive body size might be advantageous for phytotelmic frogs because it allows them to exploit smaller phytotelmata than are available to larger frogs (Scherz et al., 2019). Breeding biology of M. petrigena and M. nanapollexa is not yet reported. However, our field observations suggest that the latter species also reproduces in tree hollows. Further studies might shed light on evolutionary interdependencies between phytotelm-breeding and extreme miniaturization in Microhyla.

Taxonomic implications and cryptic diversity in Microhyla

Diminutive frogs are recognized as a source of astonishingly high undescribed cryptic diversity at different taxonomic levels due to incomplete phylogenetic information and widespread homoplasies in morphology (Hanken & Wake, 1993; Rittmeyer et al., 2012; Scherz et al., 2019). Until recently most miniature frog groups also attracted little attention by taxonomists (Rakotoarison et al., 2017). This is also true for the genus Microhyla as the only available systematic study addressing the genus by Matsui et al. (2011) provided important insights on phylogeny and taxonomy of these frogs, but did not provide other insights. In accordance with results of Matsui et al. (2011), our phylogeny indicates that morphology-based classification schemes of Parker (1934), Dubois (1987), and Fei et al. (2005) do not reflect actual phylogenetic relationships among Microhyla species, most likely due to high frequency of homoplasies both in adult and larval morphology. Further thorough morphological and osteological studies along with a robust phylogeny are required to diagnose supraspecific-level taxa within the Microhyla + Glyphoglossus assemblage.

Since the sampling used in Matsui et al. (2011) was incomplete, and the number of recognized Microhyla species has since increased almost two-fold, many questions of Microhyla taxonomy remain unaddressed. In the present article, we used an updated and almost complete phylogeny of the genus along with species delimitation methods to resolve several long-standing areas of confusion in the genus Microhyla. Our analyses suggest that despite recent progress in Microhyla taxonomy, the current number of recognized Microhyla species is still greatly underestimated. Based on species delimitation analyses, 15 (ABGD estimate) to 33 lineages (bGMYC estimate) probably reflect new species requiring formal description (Table S10). However, due to possible overlap in levels of intra- and interspecific divergence, species delineation in Microhyla based on genetic differentiation alone is problematic (Garg et al., 2019). An integrative approach including morphology and acoustics must be applied for further progress in Microhyla taxonomy (Rakotoarison et al., 2017).

Below we give a brief summary of taxonomic implications of our results. The smallest member of the genus, M. nepenthicola, was described by Das & Haas (2010) from Kubah, Sarawak, but was synonymized with M. borneensis by Matsui (2011) based on morphological examination of the M. borneensis holotype from Kidi (sic for Bidi), Sarawak, Borneo (Parker, 1928). Matsui (2011) did not include in his molecular analysis materials from the type locality of M. borneensis; however, his taxonomy was widely accepted (Frost, 2020). We analyzed topotypic M. borneensis specimen from the Bidi region (Fig. 4), specifically, from the Deded Krian National Park, near Bau, western Sarawak, and show it be a sister species of M. nepenthicola, and sufficiently divergent from the latter in 16S rRNA sequences (p = 4.6%, Table S9) to constitute a separate species. The bGMYC analysis also supports distinctiveness of M. nepenthicola from M. borneensis (Fig. 4). The name-bearing population from the former locality (M. nepenthicola) is found on sandstone massifs of western Sarawak, and is diagnosable in showing a pale brown dorsum with darker subtriangular pattern on scapular region, its adjacent areas lacking dark variegation; and flanks with an elongated dark area. On the other hand, the latter population (M. borneensis), restricted to the limestone hills of the interior, shows a gray-brown dorsum, areas outside dark subtriangular pattern with dark gray variegation; and flanks with isolated dark blotches. Therefore, we propose to revalidate the species M. nepenticola Das & Haas, 2010. Our data further suggest that M. borneensis, M. nepenthicola, and Microhyla sp. 1 from Sabah form a clade of morphologically similar and closely related taxa. Further studies are required to fully clarify morphological differences between M. nepenthicola and Microhyla sp. 1.

Significant genetic differentiation is revealed within several species of the M. achatina species group suggesting presence of cryptic diversity. Examples include M. malang (3 putative species: populations from Sarawak, Sabah and Kalimantan); M. orientalis (2 putative species: populations from Bali and Java; our study confirms the occurrence of M. orientalis in Java); M. mantheyi (2 putative species within Malayan Peninsula); M. achatina (2 putative species within Java); and M. gadjahmadai (2 putative species within Sumatra) (see Fig. 4). Genetically, the most diverse cryptic species complex in Microhyla is the M. heymonsi complex. Earlier studies already recognized the presence of several deeply divergent intraspecific lineages within M. heymonsi (Garg et al., 2019). Our new analyses revealed 7–8 highly divergent (p > 3.0%) lineages from China and northern Vietnam, Thailand and Laos, Thailand and south Vietnam, Taiwan, Myanmar, peninsular Malaysia, and Sumatra (Fig. 4). The taxonomic status of these lineages has yet to be assessed. Our study also confirms the clear distinctiveness of an undescribed species Microhyla sp. 4 from northern Myanmar, and the full species status of M. minuta, M. pineticola, M. irrawaddy, and M. fodiens, respectively.

Within the M. fissipes species group, the bGMYC analysis indicated the presence of three cryptic species-level lineages within M. mukhlesuri, although the ABGD analysis lumped M. mukhlesuri with M. fissipes (Fig. 4). Significant genetic differentiation in mtDNA-markers was reported for M. mukhlesuri by Yuan et al. (2016), but they were not corroborated by nuclear DNA. Further integrative studies are required to assess variation within M. mukhlesuri. Deep divergence was revealed between populations of M. okinavensis from Okinawa and Amami islands (p = 4.8%); populations from Yaeyama Archipelago formerly treated as M. okinavensis grouped with M. mixtura and most likely represent an undescribed species, Microhyla sp. 3 (Tominaga et al., 2019; Hasan et al., 2014a). Shallow divergence was also found among populations of M. chakrapanii from different islands of the Andaman Archipelago (Fig. 4).

Substantial genetic divergences also uncovered cryptic species lineages within the M. berdmorei species complex (Fig. 5) (Hasan et al., 2012). The bGMYC analysis suggested presence of four distinct groups from Malayan Peninsula, Malaysia + Sumatra + Borneo, Indochina, and Bangladesh. Populations from northern Thailand previously described as M. fowleri are grouped within the Indochinese lineage of M. berdmorei, suggesting synonymy of the former (see Matsui et al., 2011). Further studies, including examination of topotype material for M. berdmorei (Myanmar) and M. darevskii (central Vietnam), are needed to estimate taxonomic statuses of these newly revealed lineages and extent of their distribution. Within the M. superciliaris group, our study confirmed occurrence of M. superciliaris in southern Thailand (Songkhla), however the deep divergence of this population from the M. superciliaris population in Malaysia (p = 1.6%) suggests that it is necessary to reevaluate the taxonomy of Thai populations (Fig. 5). We also report an undescribed species Microhyla sp. 2, occurring in western Thailand where genetic variation among examined populations (Suratthani and Phetchaburi) also suggests presence of two cryptic lineages.

For the M. butleri species group, our work confirms deep divergence and full species status of the recently described M. aurantiventris (Nguyen et al., 2019), but also reveals additional undescribed lineages within M. butleri that we treat here as a species complex. From two (ABGD) to four (bGMYC) cryptic lineages were recovered within this complex, with the most divergent lineages being distributed in Taiwan + mainland China versus the rest of the species range in Indochina and the Malay Peninsula (Fig. 5). We included only two M. palmipes samples in our analysis that were notably divergent in 16S rRNA sequences (p = 3.6%) and originated from Bali and Sumatra; likely they both represent distinct species.

For the M. annectens species group, our species delimitation analyses reveal a number of cryptic and undescribed lineages. Our study confirms genetic distinctiveness of recently described M. arboricola and M. pulchella (Poyarkov et al., 2014), as well as of M. annamensis, for which genetic information was not previously available. We also added to our analysis a number of populations of M. marmorata throughout the species’ range and for the first time, including specimens of M. pulverata (collected from ca. 10 km north of the type locality in Gia Lai Province, central Vietnam). Both species were described by Bain & Nguyen (2004) based on morphological evidence and the main characters considered to be diagnostic for these species were belly coloration (marbled in M. marmorata versus dusty in M. pulverata) and skin texture. Our genetic data reveal almost no genetic differentiation between samples of M. marmorata and the topotypic M. pulverata (p = 0.4%, see Table S9), the latter are nested within the M. marmorata radiation and do not form a clade (Fig. 5). Moreover, our observations showed that belly coloration is highly variable within M. marmorata, especially in the southern part of the species range, and cannot be used as a reliable diagnostic character. Due to the lack of genetic and morphological differentiation, we hereby formally treat Microhyla pulverata Bain & Nguyen (2004) as a subjective junior synonym of Microhyla marmorata Bain & Nguyen (2004). Some other species of the M. annectens species group show deep intraspecific divergences in 16S rRNA sequences, such as M. arboricola (p = 2.6% between populations from Dak Lak and Khanh Hoa provinces of Vietnam), M. petrigena (p = 3.7% between populations from Sarawak and Sabah), and M. perparva (p = 5.1% between populations from Sarawak and Indonesian Kalimantan) (Fig. 5). It is likely that these lineages represent distinct species, including several new taxa awaiting formal description.

Conclusions

Herein, we provide an updated phylogenetic hypothesis for the genus Microhyla. An exhaustive taxonomic sampling for this group is challenging due to the high number of narrow-ranged or point-endemic species across South and Southeast Asia. In the present study, however, we examined mtDNA and nuDNA markers for 48 of 50 recognized Microhyla species (96%), including 12 nominal species and several undescribed candidate species that have not been examined phylogenetically before our work, thus providing the most comprehensive taxonomic sampling for Microhyla to date. Our data further highlight the importance of broad phylogenetic sampling and ground-level field research to gather an accurate picture of global biodiversity, phylogenetic relationships, and evolutionary patterns in cryptic groups such as microhylid frogs.

We recognize nine species groups within the Microhyla—Glyphoglossus assemblage (M. achatina, M. fissipes, M. berdmorei, M. superciliaris, M. ornata, M. butleri, M. palmipes, M. annectens species groups and Glyphoglossus), divided into three clades of probable genus-level differentiation: Microhyla I, Microhyla II and Glyphoglossus. Further integrative research combining phylogenetic and morphological lines of evidence is required to fully diagnose these recognized groups and test our taxonomic arrangement. The basal radiation of the Microhyla—Glyphoglossus assemblage is dated to the middle Eocene and likely took place in Southeast Asia. Following drifting of the Indian Plate northwards and formation of firm land bridges between the subcontinent and Asian mainland in Oligocene, ancestors of Microhyla colonized India several times from Southeast Asia and later diversified there. Our analysis also suggests that such dispersal occurred independently in five different species groups of the Microhyla II clade. Our results further corroborate the growing set of evidence for early-Eocene land connections between the Indian Subcontinent and Southeast Asia. Progressing aridification since the late Oligocene—Miocene likely facilitated dispersal of Southeast Asian biotic elements to India including the ancestral lineage-genus Microhyla. Our study further highlights the importance of Indochina not only as a cradle of autochthonous amphibian diversity and a key evolutionary hotspot for the herpetofauna (Bain & Hurley, 2011; Geissler et al., 2015; De Bruyn et al., 2014), but also as a stepping stone facilitating dispersal between Sundaland, the Indian Subcontinent, and East Asia (Woodruff, 2010; Chen et al., 2017; Suwannapoom et al., 2018; Poyarkov et al., 2018b). Further phylogenetic studies across different faunal groups with Indo-Southeast Asian affinities are required to clarify impact of complex paleogeography and paleoclimate history on formation of extant biodiversity in Asia.

Comprising the smallest tetrapods in Asia, frogs in the genus Microhyla represent a potential model group to study evolutionary drivers and constraints of vertebrate miniaturization. Our study suggests that four groups of Microhyla independently achieved extreme miniaturization with adult body sizes <15 mm. Evolution of body size in Microhyla—Glyphoglossus assemblage seems to be driven by natural history: the largest body sizes are observed in burrowing species adapted to estivation during the dry season, while three of the five smallest known Microhyla species occur only in perhumid montane forests and are phytotelm-breeders. Further research is required on how reproductive ecology in phytotelmata, often leading to reduction of clutch size, facilitates extreme miniaturization in Microhyla.

The present work clearly indicates a vast underestimation of diversity and species richness of Microhyla. We revalidate M. nepenthicola as a valid species, synonymize M. pulverata with M. marmorata, confirm species-level differentiation for a number of taxa not included in earlier phylogenies, and reveal a large number of cryptic lineages representing putative undescribed species. Alternative approaches to species delimitation suggest that at least 15–33 lineages of Microhyla likely correspond to species-level differentiation. Further integrative studies combining genetic, morphological, and acoustic parameters are essential for a better understanding of evolutionary relationships and taxonomy within this morphologically cryptic and diverse radiation of Asian frogs.

Supplemental Information

Supplemental Information 1 Geographic sampling in the present study.

Pink shading corresponds to Microhyla distribution; red circles denote localities of samples for which sequences were available via GenBank; green circles denote localities of samples for which sequences were generated in this study. For locality information see Table S1. Base Map created using simplemappr.net.

Click here for additional data file.

Supplemental Information 2 Updated mtDNA-genealogy of the Microhyla – Glyphoglossus assemblage (collapsed tree).

BI genealogy of Microhyla and Glyphoglossus reconstructed from 2478 bp of mtDNA fragment. Values at nodes correspond to BI PP/ML BS, respectively; black and white circles correspond to well-supported (BI PP ≥ 0.95; ML BS ≥ 90) and moderately supported (0.95 > BI PP ≥ 0.90; 90 > ML BS ≥ 75) nodes, respectively; no circles indicate unsupported nodes. Color marking of species groups in Microhyla species complex corresponds to Figs. 4 and 5, but not to Fig. 2. Photos by Nikolay A. Poyarkov, Indraneil Das, Vladislav A. Gorin, Parinya Pawangkhanant, Luan Thanh Nguyen, and Evgeniya N. Solovyeva.

Click here for additional data file.

Supplemental Information 3 Nuclear allele median-joining network showing relationships among phased nuclear BDNF gene haplotypes representing 49 Microhyla and Glyphoglossus species.

Circle sizes are proportional to the number of haplotypes, circle numbers correspond to sample numbers summarized in Supplementary Table S1, circle colors depict the recognized species groups, small black circles represent hypothetical median vectors, vertical bars on branches represent the number of mutational steps.

Click here for additional data file.

Supplemental Information 4 Bayesian chronogram resulted from *BEAST analysis of the 3207 bp-long concatenated mtDNA + nuclear DNA dataset.

Node values correspond to node numbers, for estimated divergence times (in Ma) see Table S11. Red icons correspond to calibration points used in molecular dating analysis, for details see Table S4. Blue bars correspond to 95%-confidence intervals.

Click here for additional data file.

Supplemental Information 5 Museum voucher information, geographic localities, and GenBank accession numbers of specimens and sequences used in this study.

Asterisk (*) denotes sequences that were included in the alignment for timetree calibration. No exact locality information is available for specimens obtained via pet trade and published in earlier works. For references see Supplementary Information file 2.

Click here for additional data file.

Supplemental Information 6 Primers used in this study.

“F,” “L”–forward primer, “R,” “H”–reverse primer. For references see Supplementary Information file 2.

Click here for additional data file.

Supplemental Information 7 The optimal evolutionary models for gene and codon partitions as estimated in PartitionFinder v1.0.1.

The optimal partitioning scheme and model fit was estimated as suggested by the Akaike information criterion (AIC).

Click here for additional data file.

Supplemental Information 8 Calibration points for divergence time estimation.

Node – tree node used for calibration, for node names see Fig. S3; divergence time given in millions years (Ma). For references see Supplementary Information file 2.

Click here for additional data file.

Supplemental Information 9 Matrix of modern species distribution within the Microhyla – Glyphoglossus assemblage.

Geographic regions: (A) Mainland East Asia; (B) Eastern Indochina; (C) Western Indochina; (D) Indian Subcontinent; (E) Malayan Peninsula; (F) Sumatra - Java - Bali; (G) Borneo and Philippines; (H) Sri Lanka; (I) East Asian Islands; see Fig. 2. No. corresponds to specimen number in Table S1.

Click here for additional data file.

Supplemental Information 10 Step-matrix showing dispersal constraints between biogeographic areas.

Four time periods correspond to: (1) 100–57 MYA marks the complete isolation of the ISC from Eurasia; (2) 57–50 MYA marks the first assumed land connections between India and the modern-day Sumatra; (3) during 50–35 MYA the ISC likely continued the counter-clockwise moving northwards forming land bridges with the modern-day Indo-Burma; and (4) the period of 35–0.0 MYA corresponds to the firm collision and formation of a stable land connection between the ISC and Eurasia. Letters encode: (A) Mainland East Asia; (B) Eastern Indochina; (C) Western Indochina; (D) Indian Subcontinent; (E) Malayan Peninsula; (F) Sumatra - Java - Bali; (G) Borneo and Philippines; (H) Sri Lanka; (I) East Asian Islands; see Fig. 2.

Click here for additional data file.

Supplemental Information 11 Body size data for the Microhyla – Glyphoglossus assemblage members.

For each species minimal, maximal and average (when available) body size data is given for both sexes. For references see File S2. Question mark denotes “no data”. Voucher IDs for specimens measured for this study are as follows: Microhyla fodiens: CAS 215851, ZMMU A5960–A5961; Microhyla irrawaddy: ZMMU A5966–A5967; ZMMU A5975–A5976; Microhyla nanapollexa: ZMMU A5635; Microhyla sp. 2: ZMMU A6032–A6035, KIZ-031270–031273; Microhyla sp. 3: ZMMU NAP-6340–6341.

Click here for additional data file.

Supplemental Information 12 Characteristics of analyzed mtDNA and nuDNA sequences.

Total length (in b.p.), number of conservative (Cons.), variable (Var.) and parsimony-informative (Pars.-Inf.) sites are given (data presented only for the ingroup).

Click here for additional data file.

Supplemental Information 13 Genetic divergence of the Microhyla – Glyphoglossus assemblage.

Uncorrected average interspecific (below diagonal) and intraspecific (on the diagonal) genetic p-distances for 16S rRNA mtDNA gene fragment (in percentage) are given for species of the Microhyla – Glyphoglossus assemblage (1–57).

Click here for additional data file.

Supplemental Information 14 Results of species delimitation analyses of Microhyla.

Number of species-level groups recovered by bGMYC and ABGD analyses presented for each of the morphospecies within Microhyla sensu lato (1–52).

Click here for additional data file.

Supplemental Information 15 Results of divergence time estimates.

Node No. – estimated tree node, for node names see Fig. S3; divergence time given in millions years (Ma).

Click here for additional data file.

Supplemental Information 16 Biogeographic area definition for South, Southeast and East Asia.

Click here for additional data file.

Supplemental Information 17 Additional references in Supplementary tables and Supplementary Information File 1.

Click here for additional data file.

Supplemental Information 18 Raw data: aligned newly generated mtDNA sequences (12S rRNA – 16S rRNA fragment).

See Supplemental Table S1 for sequence details.

Click here for additional data file.

Supplemental Information 19 Raw data: aligned newly generated nuDNA sequences (BDNF gene).

See Supplemental Table S1 for sequence details.

Click here for additional data file.

We are extremely thankful to all colleagues who helped during fieldwork, donated or facilitated obtaining samples, or discussed our hypotheses and results: S. Shonleben, V.F. Orlova, R.A. Nazarov, N.B. Ananjeva, N.L. Orlov, A.B. Vassilieva, E.A. Galoyan, D. Gabadage, M. Botejue, M. Madawala, Than Zaw, T.V. Nguyen, S.N. Nguyen, K.-H. Lee, Y.-P. Lin, H.-Y. Tseng, S.-F. Yang, O.S. Bezman-Moseiko, T. Neang, H.N. Nguyen, T. Igawa, S. Komaki, S.-M. Lin, Y. Lee, J. Lee, Y. M. Pui, K.V. Minin, A.A. Vedenin, J. Che, S. Howard, and P. Geissler. We thank A.M. Fetisova and E.S. Popov for providing important information and references on tectonics, paleogeography, and climate evolution in Asia. We are grateful to S. Shonleben, V.D. Kretova, T.V. Duong, S. Komaki, A.S. Dubrovskaya, P.V. Yushchenko, S.S. Idiiatullina, and A.N. Kuznetsov for tremendous help during our work on this project. We are also deeply grateful to Seshadri K.S., John Measey, M.D. Scherz, and an anonymous reviewer for useful comments and suggestions that helped us to improve the earlier version of the manuscript.

Additional Information and Declarations

Competing Interests

Author Contributions

Field Study Permissions

DNA Deposition

Data Availability

Nikolay A. Poyarkov is an Academic Editor for PeerJ. Suranjan Karunarathna is employed by Nature Explorations and Education Team (Sri Lanka). Luan Thanh Nguyen is employed by Asian Turtle Program—Indo-Myanmar Conservation (Vietnam). Other authors have declared that no competing interests exist.

Vladislav A. Gorin conceived and designed the experiments, performed the experiments, analyzed the data, prepared figures and/or tables, authored or reviewed drafts of the paper, and approved the final draft.

Evgeniya N. Solovyeva conceived and designed the experiments, performed the experiments, analyzed the data, prepared figures and/or tables, authored or reviewed drafts of the paper, and approved the final draft.

Mahmudul Hasan conceived and designed the experiments, performed the experiments, analyzed the data, authored or reviewed drafts of the paper, and approved the final draft.

Hisanori Okamiya conceived and designed the experiments, performed the experiments, analyzed the data, authored or reviewed drafts of the paper, and approved the final draft.

D.M.S. Suranjan Karunarathna analyzed the data, authored or reviewed drafts of the paper, and approved the final draft.

Parinya Pawangkhanant analyzed the data, prepared figures and/or tables, authored or reviewed drafts of the paper, and approved the final draft.

Anslem de Silva analyzed the data, authored or reviewed drafts of the paper, and approved the final draft.

Watinee Juthong analyzed the data, authored or reviewed drafts of the paper, and approved the final draft.

Konstantin D. Milto analyzed the data, prepared figures and/or tables, authored or reviewed drafts of the paper, and approved the final draft.

Luan Thanh Nguyen analyzed the data, authored or reviewed drafts of the paper, and approved the final draft.

Chatmongkon Suwannapoom performed the experiments, analyzed the data, authored or reviewed drafts of the paper, and approved the final draft.

Alexander Haas conceived and designed the experiments, performed the experiments, analyzed the data, prepared figures and/or tables, authored or reviewed drafts of the paper, and approved the final draft.

David P. Bickford analyzed the data, prepared figures and/or tables, authored or reviewed drafts of the paper, and approved the final draft.

Indraneil Das conceived and designed the experiments, performed the experiments, analyzed the data, prepared figures and/or tables, authored or reviewed drafts of the paper, and approved the final draft.

Nikolay A. Poyarkov conceived and designed the experiments, performed the experiments, analyzed the data, prepared figures and/or tables, authored or reviewed drafts of the paper, and approved the final draft.

The following information was supplied relating to field study approvals (i.e., approving body and any reference numbers):

No field studies were carried out specifically for this work; tissue samples and specimens stored in museum collections were used in this study. However, some specimens stored in the mentioned collections were collected by the coauthors of this manuscript or with their participation during numerous field trips in a time frame over 15 years. Specific permits from Malaysia, Sri Lanka, Thailand and Vietnam are as follows: - The Sarawak Forest Department and the Sarawak Forestry Corporation granted a fieldwork permit to Indraneil Das (permit number (13)JHS/NCCD/600-7/2/107(Jld2));

- The Sarawak Forest Department and the Sarawak Forestry Corporation granted permission for fieldwork under permit number NPW.907.4.2–26, NPW.907.4.2–43; NPW.907.4–35; NPW.907.4–36; NPW.907.4.2–8 and NPW.907.4.2(II)–73; NCCD.907.4.4 Jld.7–39, issued to Indraneil Das (Malaysia), and export permits (04635, 07094–97 and 07484);

- The Department of Wildlife Conservation of Sri Lanka granted fieldwork permission (permit number WL/3/2/1/14/12 issued to Anselm de Silva, 30 April 2008);

- The Institute of Animals for Scientific Purpose Development (IAD), Bangkok, Thailand granted permission for fieldwork under permit number U1-01205-2558, issued to Chatmongkon Suwannapoom;

- The Institute of Animals for Scientific Purpose Development (IAD), Bangkok, Thailand granted permission for fieldwork under permit number UP-AE59-01-04-0022, issued to Chatmongkon Suwannapoom (Thailand);

- The Institutional Ethical Committee of Animal Experimentation of the University of Phayao approved field collections under certificate number 610104022 issued to Chatmongkon Suwannapoom;

- The Department of Forestry, Ministry of Agriculture and Rural Development of Vietnam granted permission for fieldwork under permit number #170/TCLN–BTTN (issued 7 February 2013), #831/TCLN–BTTN (issued 5 June 2013), #400/TCLN-BTTN (issued 26 March 2014), #547/TCLN-BTTN (issued 21 April 2016), #432/TCLN-BTTN (issued 30 March 2017), issued to JRVTRTC, Nikolay A. Poyarkov (Vietnam);

- The Department of Forestry, Ministry of Agriculture and Rural Development of Vietnam approved additional fieldwork (permit numbers #142/SNgV-VP (Gia Lai Province, issued 4 May 2016), #1539/TCLN-DDPH (issued 19 September 2018), #1700/UBND.VX (Nghe An Province, issued 22 March 2018) and #308/SNgV-LS (Quang Nam Province, issued 01 April 2019) given to Nikolay A. Poyarkov), Thanh Hoa Province (#3532/UBND-THKH, issued 27 March 2019); Lam Dong Province (#5832/UBND-LN, issued 22 October 2012); Dak Lak Province (#1567/UBND-TH, issued 06 April 2011; #388/SNgV-LS, issued 24 April 2019; #995/SNN-CCKL, issued 12 April 2019); Lao Cai Province (#1148/UBND-TNMT, issued 26 March 2019);

- In Cat Tien NP, fieldwork was conducted in accordance with Agreement #37/HD on the scientific cooperation between Cat Tien NP and JRVTTC;

- In Bu Gia Map NP, fieldwork was conducted in accordance with Agreement #137/HD NCKH of 23 June 2010 on the scientific cooperation between Bu Gia Map NP and JRVTTC;

- Forest Protection Departments of the Peoples’ Committee of Gia Lai Province granted permission for fieldwork to Nikolay A. Poyarkov (permit number #530/UBND-NC (issued 20 March 2018)).

The following information was supplied regarding the deposition of DNA sequences:

The sequences described here are available at GenBank: MN534393 to MN534449; MN534450 to MN534550; MN534551 to MN534657; MN534658 to MN534765; MK208926 to MK208938; MH286426 to MH286427.

The following information was supplied regarding data availability:

The raw data has been supplied as voucher specimens and tissue samples stored in the following herpetological collections: Zoological Museum of Moscow University (ZMMU; Moscow, Russia);

Zoological Institute, Russian Academy of Sciences (ZISP; St. Petersburg, Russia);

Vertebrate Zoology Department, Biological Faculty, Moscow State University (ZPMSU; Moscow, Russia);

Amphibian Research Center, Hiroshima University (IABHU; Higashihiroshima, Japan);

Danum Valley Conservation Area, Specimen collection (RMBR; Sabah, Malaysia);

Department of Fisheries, Bangamata Sheikh Fazilatunnesa Mujib Science & Technology University (DFBSFMSTU; Jamalpur, Bangladesh);

School of Agriculture and Natural Resources, University of Phayao (AUP; Phayao, Thailand);

Institute of Biodiversity and Environmental Conservation, Universiti Malaysia Sarawak (UNIMAS; Sarawak, Malaysia).

For the full list of voucher specimens examined, see Table S1.

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
