# Peer review of "A little frog leaps a long way: compounded colonizations of the Indian Subcontinent discovered in the tiny Oriental frog genus Microhyla (Amphibia: Microhylidae)"

_PeerJ, doi:10.7717/peerj.9411_

## Round 0.1 · original submission · Minor Revisions

Both reviewers are of the opinion that your manuscript needs only minor revision before it can be accepted for publication.

I would like to highlight that you have opposed many authors who have published previous relevant work. In doing so, you must accept that your manuscript may not benefit from all relevant opinions while in review. I do not consider this a fair situation from the standpoint of these other authors and I think that you should reconsider barring quite so many researchers who hold most of the insight that might help your study - facilitating rigorous peer review. Otherwise, you should provide some valid explanation of exactly why you wish to bar all other interested parties.

In addition, please remove the word 'invasions' from your title as this word technically means that these animals were introduced by humans.

Reviewer 1 ·

Basic reporting

Abstract – The MS has three parts: time-calibrated phylogenetic analysis, reconstructed biogeographic histories, and morphological evolution. I think the readers would appreciate it if you add a sentence stating the aim of the study (mentioning the different parts) before presenting a summary of the results of each part.
My comments on the introduction are mainly on framing the study.
Introduction, Paragraphs 1 and 2, Lines 66–98:
Paragraph 1 framed post-collision biotic interchange between the Indian Subcontinent and Eurasian landmass; whereas, Paragraph 2 framed (mainly) pre-collision biotic interchange.

In your 2nd paragraph, you opened the paragraph with a passing statement on the diversity of the indo-malayan realm, then discussed pre-collision biotic interchange. I think these two topics, along with the 1st paragraph on post-collision biotic interchange, could be better framed with the following suggestions:
1st Paragraph – a short paragraph on the extreme diversity in the Indomalayan Realm (for amphibians, Bain et al. 2008 is a good starting resource -- found in Stuart et al. 2008 [opensource]), then perhaps enumerating key geological and climatic histories as factors that caused such diversity. Also, I suggest not using "global biodiversity hotspots" as, although it infers high biological diversity, it is more of a biodiversity conservation-linked term, rather than a biogeographic (theoretical) term.
Bain, R., Biju, S. D., Brown, R. M., Das, I., Diesmos, A. C., Dutta, S., ... & Lau, M. W. N. (2008). Amphibians of the Indomalayan realm. Threatened Amphibians of the World. Lynx Ediciones,
Stuart, S. N. (Ed.). (2008). Threatened amphibians of the world. Lynx Edicions.

2nd and 3rd (or into one paragraph) Paragraphs – Frame the competing biogeographic hypotheses: you identified four potentially competing hypotheses – (1) post-collision “out-of-India”, (2) post-collision “out-of-Eurasia”, (3) pre-collision “out-of-India, (4) pre-collision “out-of-Eurasia”.
Few notes,
Introduction, Line 79: the use of “however” as conjunction infers the "out-of-India" and "out-of-Eurasia" hypotheses as competing (mutually exclusive) hypothesis. However, the fact that multiple evidence supports both hypotheses suggests non-unidirectional biotic interchange -- both may have co-occurred. "However" may not be the most appropriate conjunction.

Paragraphs 4 and 5, Lines 115–144 framed the genus Microhyla in terms of its known diversity, distribution, and phylogeny.
Lines 121–123: “Microhyla was questioned by …” omit sentence from the paragraph; I think it is better placed into the next paragraph.
Line 127: “muddying” – use another term.
Line 120: Cite the source of the geographic range maps used in creating the figure (e.g., adopted from a previous publication, IUCN, expert delimited, etc.).
Paragraph 5, Lines 132–144: frames the rationale of the study. This paragraph discusses the current knowledge of Microhyla phylogenetics and biogeography.
I recommend more synthesis of thought – (i) start with an overarching opening sentence (that covers phylogenetics and biogeographic inferences (rather than having both topics discussed separately within the paragraph). Else, break this paragraph to two: one for phylogeny and another for biogeography. (ii) synthesize (and give more justice) to literature -- rather than simply describing them in passing. (iii) lastly, enumerate the shortfalls or aspects where we can further advance our understanding of the froup’s phylogeny and biogeography.
There will definitely be some redundancy with parts of your discussion, but I think this is inevitable; I think it is important for you to show your readers that you have thoroughly dissected the current literature (especially that latest studies on Microhyla phylogenetics was very recent), identified shortfalls and areas of advancement. I agree that one of the major gaps in past research is the biased taxonomic/geographic representation of species (mostly assessed from the Indian Subcontinent), and that wider taxonomic sampling will result in phylogenetic trees and biogeographic inferences at finer resolutions – as what you have done in your study; leverage on this.

Lines 134–137: “However, they were … “ – break and rephrase this sentence.

Paragraph 6, Lines 145–155: the aims of the study.
Lines 147–148: “… ; phylogenetic information for 12 species and a number of undescribed species … “ make sure to explicitly express that these are presumably undescribed species.

Materials and methods
Taxon sampling

Lines 159–168 – make sure to cite at the end of this paragraph Table S1.
Lines 175–176 – Add a supplementary figure of distribution of samples; analogous to Fig S1 but delimited to species (instead of sampled and genbank) and with each point labeled accordingly (i.e., numbering of samples used in the MS). This gives a better picture of your taxonomic sampling, particularly for species with wide and disjunct ranges (putatively comprising of multiple sub/species – as shown in your study).

Lines 177 – “… removed …” – sampled?


Results
phylogenetic relationships, and species groups in Microhyla.
- For brevity, and a matter of taste: in describing the distribution of species (or clades), maybe use a consistent geographical terminology; septically, I suggest consistently describing geographic distribution in the context of biogeographic areas (as enumerated in Figure 2 and described in Supp Info 1), rather than geopolitical regions (e.g., Myanmar, India, China, etc.).

Discussion, same comment with results.

Figure 1 – as mentioned above, cite the source of data on species geographic ranges. also cite the software used to create the maps.

Figure 4–7 – If you can follow one single theme for both figures; also, although I highly appreciate images of representative species (it definitely enables readers to better appreciate the size variations in the group), I think it somewhat clutters the figures (overwhelming) in some instances. I suggest selecting a good representative species for each clade (one that exemplifies the morphology of the clade), if possible. If it is not yet scaled in its current form, scale the figures of species by their size and include a scale bar.

Legends of the supplementary tables and figures were not supplied.

Add a supplementary figures/table detailing the results of the phylogenetic analysis on proximal nodes of the trees (intra-specific) – although may not be informative as the markers used may not be able to capture inter-population genetic variations. Detailed reconstruction of the phylogeny of numerous amphibian groups in the Indomalayan realm (e.g., Polypedates leucomystax – Brown et al. 2010; Duttaphrynus melanostictus – Reilly et al. 2017) revealed fairly recent inter-population admixtures, as a result of human-mediated dispersal and introduction, and biogeographers and invasion biologists speculate genetic admixtures to be common as well in other widely distributed amphibian groups (and other taxa) in the region. Microhyla is likely one of these amphibian groups; three Microhyla spp. (M. fissipes, M. pulchra, M. ornata, and potentially more) have been introduced (recently) and/or established outside their native ranges, and as mentioned in Lines 713–715, human-mediated dispersal and introduction have been previously hypothesized in M. kodial (but disproved by this study). Providing information on the findings of Phylogenetic analysis on the proximal nodes of the trees would be useful for future researchers interested in recent-human-mediated colonization events.

Experimental design

As mentioned in the previous section, give more context on current lit in the introduction.
In Supplementary Table S7, identify samples (you) measured in the study (for morphological evolution) and their corresponding accession numbers, if any. You did mention that you gathered data from the literature, and measured museum specimens.


For research ethics purposes, cite (in acknowledgment), where applicable, permits acquired to conduct the field sampling.

Validity of the findings

Overall, the primary contribution of their MS is the phylogenetic analysis, which has an almost complete representation of currently recognized (in addition to potentially undescribed) species and thereby, with a finer resolution. Secondly, with their time-calibrated phylogenetic analysis, the provided biogeographic inference that is arguably more robust than recent past studies; their overall findings on the biogeography of Microhyla conforms with previous studies – out-of-Asia colonization to Indian sub-continent. Microhyla thereby in all technicality supports the post-collision out of Asia hypothesis, but, as also a case of Microhylinae recolonization, does it really? Or is it strong evidence in support of the hypothesis?

Additional comments

It is a pleasure to be given the opportunity to read and comment on the manuscript (hereafter “MS”) of Gorin et al., which updated the current knowledge on the phylogeny of the genus Microhyla (family Microhylidae; with dating estimates, and a near-complete representation of currently recognized distinct species), reconstructed the group's biogeographic history and diversification, and analyzed their morphological evolution in the context of miniaturization

I am convinced that the MS is a significant contribution to the growing body of knowledge on faunal biogeography in the indo-malayan realm, and particularly to a cryptic and widely distributed group of amphibian(s) – Microhyla spp. – exhibiting a diverse, peculiar, and novel eco-morphological evolution.
The MS was well-written (great technical quality), its theoretical framework sufficiently contextualized (but can be further improved), and findings were well-described and discussed; however, the MS can further improve in few minor aspects on basic reporting and experimental design.

·

Basic reporting

I found the MS titled “A little frog leaps a long way: Multiple invasions to the Indian
Subcontinent discovered in the Oriental tiny frog genus Microhyla (Amphibia: Microhylidae)” to be an interesting read. It provides an insight into the origins and biogeography of frogs in the genus Microhyla using molecular evidence.

The MS is clearly written, referenced with up to date citations and placed in the broad context of what is known about Microhyla biogeography. The work is relevant and an important contribution to advancing the field and the narrative throughout the MS is cogent and self-contained.

A minor issue I had was with how the results were written. I have marked out sections of the results which need to be re-written because the authors simply point the reader to a table or a figure in the results by saying- results of this analysis are shown in fig xx. As a reader, I expect to read your results/interpretations and not see a figure and understand things from it. Figures and tables should be stand-alone but they should also complement your text.

Experimental design

The authors have sampled to include most of the species in the genus they are studying and use relevant resources already available as well. The raw data is provided and accession numbers to resources used from literature are provided. The methods provided are in sufficient detail to understand and even repeat the study.

Validity of the findings

The results from this paper are insightful and such a study was long overdue. In recent years, many species of Microhyla were described and it was only a matter of time till someone tested the hypothesis of their evolutionary origins. This MS bridges a large knowledge gap by providing an updated phylogeny, delimitation of potential undescribed lineages, and providing evidence to a SE Asian origin hypothesis.

Additional comments

I enjoyed reading your MS but struggled to easily grasp the details largely because of how results were presented and having had to go back and forth from one figure to another. I think your figures are good but contain too much information and need to be explained for e.g. see sections of the discussion highlighted in the PDF. With a little tweaking of how results are conveyed- I think the paper will be widely read and comprehended easily.

---

## Round 0.2 · accepted · Accept

Thanks for your revision, and I apologise for the delay in getting this decision back to you. In part a problem I had with email, but also a late review that we were chasing.

Please note that reviewer 1 has some outstanding minor corrections that are still required, but I think that these can be picked up in copy editing. You are welcome to keep your phylogenetic comparisons in the results as is traditional.

Reviewer 1 ·

Basic reporting

firstly, let me respond to their response to my comments/suggestions on the earlier version of the manuscript:
Regarding comment REV1C9, I meant proper citation of spatial data: see the “Citing Spatial Data” section in https://www.iucnredlist.org/about/citationinfo
Examples of full citations for spatial data:
IUCN (International Union for Conservation of Nature) 2019. Ochotona iliensis. The IUCN Red List of Threatened Species. Version 2020-1. http://www.iucnredlist.org. Downloaded on 19 March 2020.

I suggest further proofreading.

Abstract, Line 33:
Omit “The small-sized genus Microhyla” as this (its small size) has already been mentioned in the previous sentence. Replace this with “The genus”.

Abstract, Line 37–39:
“Our updated phylogeny of the genus with nearly complete taxon sampling includes 48 nominal Microhyla species and several undescribed candidate species.” This doesn’t sound right. Maybe try:
“Our updated phylogeny of the genus includes 48 nominal species and several undescribed candidate species, constituting a nearly complete taxon sampling.”

Abstract, Line 39:
“… both …” Both or combined?

Abstract, Lines 39–42:
“Phylogenetic analyses of 3207 bp of both mtDNA and nuDNA data recovered three well-supported groups: the Glyphoglossus clade, Southeast Asian Microhyla II clade (includes M. annectens species group), and a diverse Microhyla I clade including all other species.”
I have mixed opinions on the inclusion of “Glyphoglossus clade” here; wouldn’t it be more sound (with regards having a clear focus) to say instead “… supports monophyly in the genus Microhyla, with genus Gluphoglossus as its sister taxa. Moreover, our analysis recovered two well-supported groups within the genus: Microhylia I clade … ” Or something like that.

Introduction, Lines 73–76:
Suggested edit: “Notably, recent studies combining data on the region's tectonic history and paleoclimate with phylogenetics of different faunal and floral groups (e.g., De Bruyn et al., 2014; site ~2 more landmark papers) has enabled thorough examination of the biogeographic and evolutionary processes that ultimately lead to the region's high biodiversity.” Or something like that.

Introduction, Line 86:
“…) the so-called, ..”
Suggested edit: “; this biogeographic hypothesis has been referred to as …”

Introduction, Line 89:
“…animal…” ; be consistent in using: plant and animal vs. floral and faunal.

Introduction, Line 91:
“… during its collision”; during or post-collision?

Introduction, Line 94–95:
“There is an ongoing debate on timing and topography of the ISC–Eurasian collision … ” The body of the paragraph does not relate to its opening. I suggest omitting the first sentence or transferring it to the previous paragraph.

Introduction, Line 100:
omit “demonstrating that they likely went in both directions: “out of India” and “out of Eurasia”; I suggest replacing this with “and suggested that the faunal exchange between regions went in both directions.”

Introduction, Line 102 – 104:
Suggested edit: “Despite recent advancements in our understanding of the biogeographic history of the region, the implications of pre- and post-collision “out-of-India” or “out-of-Eurasia” biogeographic scenarios on biotic exchanges between the ISC and the Asian mainland remains unresolved.”

Introduction, Line 121:
I suggest the following edit: “The genus Microhyla is the only Asian microhylid genus with a wide distribution over South, Southeast, and East Asia (see Fig. 1). It is also the most species-rich genus in the Microhylinae, currently comprising 50 recognized species (Poyarkov et al., 2014, 2019; Biju et al., 2019). Because of its wide distribution and high species diversity, It is an ideal model system for studies on Asian biogeography.”

Introduction, Lines 130–133:
“Molecular phylogenetic …”
I have mixed opinions on this sentence: You identified here a problem and a possible solution, but this isn’t exactly something addressed by your study (although iteratively repeated in the discussion); you only fulfilled one part of this herculean task –molecular phylogenetic analysis with a near-complete taxon sampling. You also elaborated gaps in knowledge of the species’ molecular phylogeny in the next paragraph.
Maybe omit this?

Introduction, Line 134–137:
I suggest omitting the first two sentences (the latter, I suggest transferring to the previous paragraph). Begin with “Phylogenetic relationships …”

Introduction, Line 150–151:
“… molecularly previously unstudied …” doesn’t sound right.

Introduction, Lines 161:
maybe omit “the first”

Introduction, Lines 161 – 163:
This is not clear, can you revise this (especially the “links the Indian Subcontinent with Sundaland, Indo-Burma, and East Asia….”).

Materials and Methods, Lines 178–187:
Although I applaud that the field surveys (not yours) from which the specimens were collected were conducted ethically, I suggest transferring the listed permit numbers (and granting institutions) in the acknowledgement. You can simply write here: “Permits to conduct fieldwork and collect specimens, where applicable, were obtained from respective institutions.” Or something like that.

Materials and Methods, Line 327:
“… the tremendous geological complexity of the region through time,”
Not sure if this is correct: what about "Given the region's highly complex geological history"?

Discussion, Lines 550–566:
I suggest, prior to discussing your findings (updated phylogeny), give context first on earlier studies on Microhyla phylogeny. Thus, I suggest transferring Lines 551–556 (starting from “we present …” and as a paragraph) after paragraph Lines 579–591. Confer to how you discussed biogeography.

Discussion, Line 592:
Prior to discussing the individual species groups, is it possible to discuss very briefly the two major clades (Microhyla I and II), particularly their morphological differences.

Experimental design

Background on relevant theories and concepts are well contextualized. Relevant literature well and succinctly reviewed (in the introduction). Knowledge gaps and their implications were identified. Methods were sufficiently detailed, allowing replicability of the study. Their extensive supplementary materials detailing the materials used is a plus. The analysis (and data collection) were performed to a high ethical standard.

Validity of the findings

The conclusions addressed their original research question and were consistent with/limited to their findings. Recommendations for future research is a plus. The plausibility of their speculations was sufficiently argued, evidence-based, and logical. Underlying data shared as supplementary materials is a plus.

Additional comments

It was an absolute pleasure to be given the opportunity to comment to and provide suggestions for the improvement of Gorin et al.’s manuscript. It is good to know that they considered both reviewers comments; for my comments and suggestions, I am very satisfied with their point-by-responses. Overall, the manuscript significantly improved.
I commend how they thoroughly edited/revised the results and discussion section (not to mention extending their analysis). It is much clear now how their paper stacked-up to related literature (especially to very recent papers on the topic), and how it advanced our understanding on Microhylidae/Microhyla phylogenetics and biogeography, and Asian biogeography in general.
Although I have no more comments on the experimental design and the validity of the findings, I think the manuscript can still benefit with a few improvements on basic reporting.
1. Introduction section. The background on theories and concepts have significantly improved, and current state of knowledge on Microhylidae and Microhyla phylogenetics sufficiently contextualized; however, I think the manuscript can benefit with some further improvement on the construction and flow of thought. I also suggest further proofreading. I enumerated my suggestions in the basic reporting section.
2. Results section. I suggest limiting (if not omitting any) comparisons here (i.e., comparing your findings with previous studies, whether they conform or vary). I suggest strictly describing your findings here and keep your comparisons in the Discussion section. You did mention in your response that you wanted to keep the results concise; I think making comparisons here does the contrary.
3. Discussion section. Overall, the discussion section is superb! I just have a minor suggestion (see the basic reporting section).
4. Figures. I am satisfied with your response to my comment/suggestion on Figure 1. For the other figures (except figure 3), It’s important to note that both reviewers had similar comments on them (i.e., that some figures are a bit overwhelming). There were hardly any changes/improvements in the figures.

·

Basic reporting

The manuscript has been revised sufficiently. The MS now has increased clarity and flow throughout.

Experimental design

It is nice of the authors to include the information on taxon sampling and permits.

Validity of the findings

The authors have improved the analysis and this provides more basis for the conclusions they are drawing. The discussion section is substantially improved as well.

Additional comments

None.